# Weakly coordinated Li ion in single-ion-conductor-based composite enabling low electrolyte content Li-metal batteries

Hyeokjin Kwon [1], Hyun-Ji Choi[1], Jung-kyu Jang [2], Jinhong Lee[1], Jinkwan Jung[1], Wonjun Lee[3], Youngil Roh[1], Jaewon Baek[1], Dong Jae Shin[1], Ju-Hyuk Lee[1], Nam-Soon Choi [1] ✉, Ying Shirley Meng[4,5] ✉ & Hee-Tak Kim [1,6] ✉

The pulverization of lithium metal electrodes during cycling recently has been suppressed through various techniques, but the issue of irreversible consumption of the electrolyte remains a critical challenge, hindering the progress of energy-dense lithium metal batteries. Here, we design a single-ion-conductor-based composite layer on the lithium metal electrode, which significantly reduces the liquid electrolyte loss via adjusting the solvation environment of moving $Li^+$ in the layer. A $Li||Ni_{0.5}Mn_{0.3}Co_{0.2}O_2$ pouch cell with a thin lithium metal (N/P of 2.15), high loading cathode (21.5 mg cm$^{-2}$), and carbonate electrolyte achieves 400 cycles at the electrolyte to capacity ratio of 2.15 g Ah$^{-1}$ (2.44 g Ah$^{-1}$ including mass of composite layer) or 100 cycles at 1.28 g Ah$^{-1}$ (1.57 g Ah$^{-1}$ including mass of composite layer) under a stack pressure of 280 kPa (0.2 C charge with a constant voltage charge at 4.3 V to 0.05 C and 1.0 C discharge within a voltage window of 4.3 V to 3.0 V). The rational design of the single-ion-conductor-based composite layer demonstrated in this work provides a way forward for constructing energy-dense rechargeable lithium metal batteries with minimal electrolyte content.

Since their inception in the 1970s[1], lithium-ion batteries (LIBs) have evolved greatly, influencing modern society and industry. However, with amplifying demand for the greater energy density of batteries in electric vehicle applications, lithium (Li) metal has been revisited as a negative electrode as the capability of state-of-the-art LIBs approaches the theoretical limit[2–4]. Unfortunately, Li metal has a critical weakness: the chemical and electrochemical instability of the lithium/electrolyte interface causes the formation of a thick byproduct layer, sluggish ion transport, and continued consumption of the Li inventory and residual electrolyte[3,5–7]. The poor levels of reversibility of Li metal batteries (LMBs) have led to the use of a thick Li anode as a Li reservoir, which substantially decreases the volumetric energy density of packaged LMB cells, diminishing the advantages of LMBs.

Although the reversibility of Li metal electrodes has been notably improved by various strategies including electrolyte engineering[8–11], the use of an artificial solid electrolyte interphase (SEI)[12,13], and the implemention of a protective layer[14–17], achieving a durable and energy-dense LMB with low electrolyte content (<3 g Ah$^{-1}$) remains challenging. According to various diagnostic analyses reported in this field, it is expected that this challenge originates from the lack of a strategy to radically halt the decomposition of the electrolyte on the Li surface[18,19]. Recent studies have shown that molecules coordinating

[1]Department of Chemical and Biomolecular Engineering, Korea Advanced Institute of Science and Technology (KAIST), Daejeon, Republic of Korea. [2]Energy Materials Research Center, Korea Research Institute of Chemical Technology (KRICT), Daejeon, Republic of Korea. [3]Department of Energy Engineering, School of Energy and Chemical Engineering, Ulsan National Institute of Science and Technology (UNIST), Ulsan, Republic of Korea. [4]Department of NanoEngineering, University of California at San Diego, San Diego 92093 CA, USA. [5]Pritzker School of Molecular Engineering, University of Chicago, Chicago, IL, USA. [6]Advanced Battery Center, KAIST Institute for the NanoCentury, Korea Advanced Institute of Science and Technology (KAIST), Daejeon, Republic of Korea. ✉e-mail: nschoi@kaist.ac.kr; shirleymeng@uchicago.edu; heetak.kim@kaist.ac.kr

with Li⁺ are more likely to decompose on the Li surface; due to the migration of solvated Li⁺ to the Li metal electrode, the coordinated molecules are concentrated at the Li surface and decomposed by Li metal with a higher probability[17,20,21]. Therefore, along with densification of the Li morphology, engineering modifications of the Li⁺ solvation structure at the Li-electrolyte interface to minimize electrolyte decomposition are required to further advance the performance of energy-dense LMBs. In contrast with the efforts to date to generate a robust SEI structure by selecting reactive electrolyte components and adjusting their composition[22–24], we focus on the weak coordination of electrolyte molecules from Li⁺ to address the aforementioned reactivity between Li and liquid electrolyte.

Here, we present a composite layer of single-ion-conducting ceramic electrolyte (S-CE) and single-ion-conducting polymer (S-PE)-based gel electrolyte (S-GE) coated on a Li metal electrode as a modulator of the reactivity of liquid electrolyte components. We reveal that the composite layer of S-CE and S-GE (S-CE/S-GE) reduces the loss of liquid electrolyte on the Li metal electrode and improves the cycling stability of LMBs by changing the solvation environment of Li⁺ in the layer (Fig. 1a); this effect is not seen with a composite layer comprising S-CE and a bi-ion-conducting polymer (B-PE)-based gel electrolyte (B-GE). Furthermore, we demonstrate that the S-CE/S-GE layer affects the morphology of Li anodes and the stability of the cathode–electrolyte interface in Li metal batteries. Exploiting the S-CE/S-GE composite layer, we developed an energy-dense Li metal pouch cell (Li|S-CE/S-GE|| LiNi$_{0.5}$Mn$_{0.3}$Co$_{0.2}$O$_2$) and demonstrated operation for over 400 cycles with low mass of electrolyte content of 2.15 g Ah⁻¹ (2.44 g Ah⁻¹ including mass of composite layer) or 100 cycles at 1.28 g Ah⁻¹ (1.57 g Ah⁻¹ including mass of composite layer) even in typical carbonate electrolyte. The judicious engineering of the solvation structure and

consequent results provide new insights into the control of interfacial reactions and a practical methodology to improve LMBs.

## Results

### Construction of the S-CE/S-GE composite layer

It is well known that when two single-ion-conductive phases are in contact, a space charge layer is formed due to the difference in the concentration (i.e., chemical potential) of Li⁺ in the two phases[25,26]. We anticipated that the Li⁺ space charge generated in the S-CE/S-GE composite layer will have an abnormal Li⁺ coordination structure, and investigated how the coordination structure of Li⁺ in the S-CE/S-GE composite layer affects the behavior of liquid electrolyte decomposition. For this purpose, we employed Li$_{6.4}$La$_3$Zr$_{1.4}$Ta$_{0.6}$O$_{12}$ (LLZTO) as a S-CE and poly(styrene trifluoromethanesulphonylimide)lithium (P(STFSI)Li)-co- Poly(ethylene glycol) diacrylate (PEGDA)[27] that is a crosslinked single-ion-conducting polymer electrolyte (S-PE) as a matrix for S-GE (Supplementary Fig. 1). The model composite layer consisting of the S-CE and S-PE, that is denoted as S-CE/S-PE was formed on a Li metal electrode surface by tape-casting and subsequent polymerization of a slurry mixture of LLZTO particles, (STFSI)Li monomer, and PEGDA (Fig. 1b and Supplementary Fig. 2). It is converted to S-CE/S-GE in contact with liquid electrolyte (Supplementary Fig. 3). To rule out the effects of reduced contact between the Li electrode and the liquid electrolyte by the ceramic and the polymer, or other incidental variables that occur in the slurry coating process, we compared the S-CE/S-GE with a composite layer with a bi-ion conducting polymer (B-PE)-based gel electrolyte (denoted as B-GE) consisting of lithium bis(trifluoromethanesulfonyl)amide (LiTFSI) and PEGDA (see method section). The two layers, which were prepared in the same manner, both reduce the contact between the Li electrode

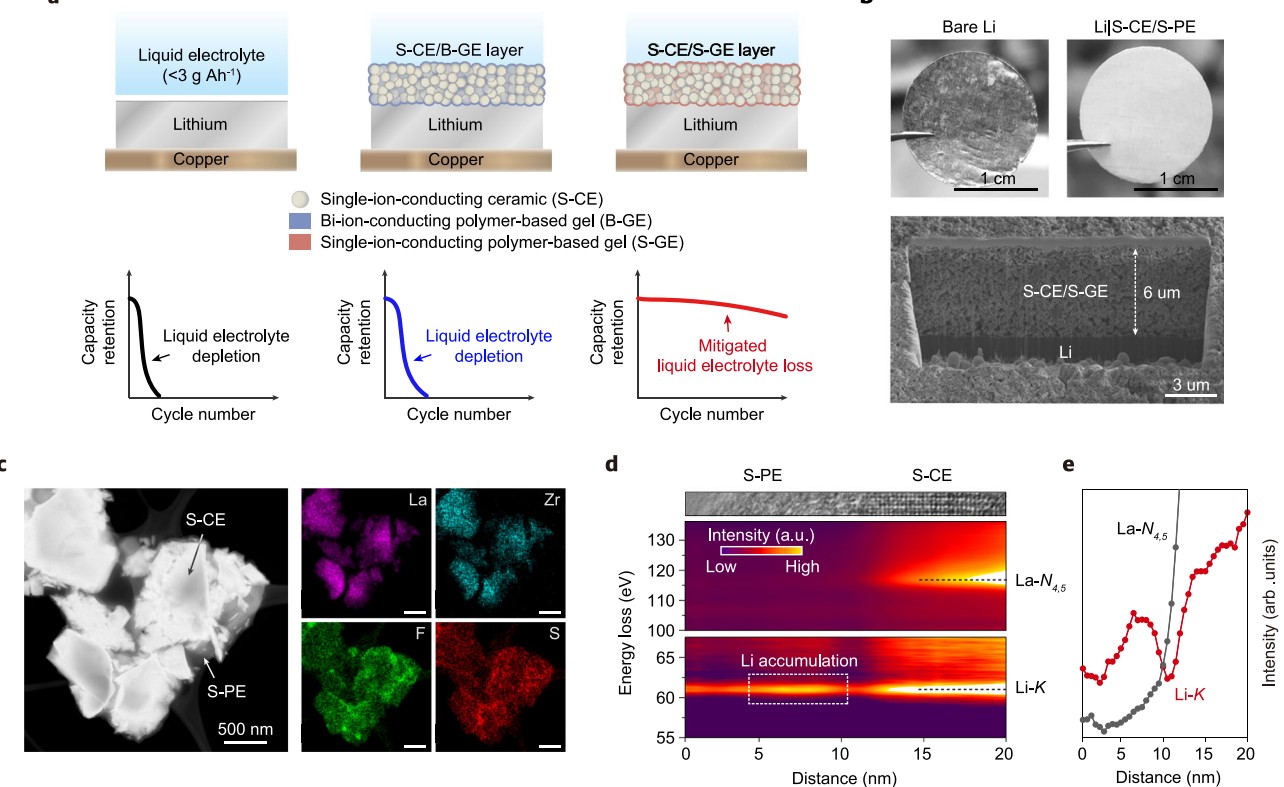

**Fig. 1 | Single-ion-conductor-based composite layer. a** Schematic illustration of the effects of the S-CE/B-GE and S-CE/S-GE composite layers on the cycling stability of liquid electrolyte-based LMBs. **b** Optic and FIB-SEM images of S-CE/S-PE composite layer coated on Li metal electrode. **c** STEM-EDX mapping of S-CE/S-PE composite. Scale bar: 500 nm. **d** TEM image and SR-EELS image recorded with scanning across the S-CE/S-PE interface at the Li-$K$ (55–70 eV) and La-$N_{4,5}$ (100–140 eV) energy loss region. **e** The corresponding intensity profiles of Li-$K$ and La-$N_{4,5}$.

and the liquid electrolyte. It should be noted that the Li⁺ transference number is 0.61 for S-CE/B-PE and 0.96 for S-CE/S-PE, indicating a difference in ion transport property due to the polymer structure (Supplementary Fig. 4).

The heating temperature producing the composite layer was set to 60 °C to ensure uniformity of the composite layer (minimizing cracking or pinholes, Supplementary Fig. 5). The amount of the residual slurry solvent in the composite layer was only 0.0013 g Ah⁻¹ (0.9 wt% in polymer, Supplementary Fig. 6) based on the cathode capacity (3.72 mAh cm⁻²) designed later; the amount of the residual solvent could be ignored compared with the amount of the liquid electrolyte injected (1–2 g Ah⁻¹).

The thickness of the S-CE/S-PE composite layer was controlled to be in a range of 6.0–7.0 µm, as shown in the scanning electron microscopy (SEM) images (Fig. 1b). The XRD and selected area electrode diffraction pattern of the S-CE/S-PE composite indicate that LLZTO particles (S-CE) maintained their original phase with the Ia-3d space group after fabrication of the composite layer (Supplementary Fig. 7). The S-CE and S-PE formed conformal contact as verified by the results of a scanning transmission electron microscopy (STEM)-energy dispersive spectroscopy (EDX) analysis for a mixture of the S-CE and S-PE (Fig. 1c); the lanthanum and zirconium EDX areas from the S-CE are within the fluorine and sulfur EDX areas from the S-PE, indicating that the S-PE surrounds the S-CE particles.

To directly observe the Li⁺ space charge at the interface between the S-CE and S-PE, the Li⁺ distribution across the S-CE/S-PE interface was quantified using spatially resolved electron energy-loss spectroscopy (SR-EELS). Figure 1d, e presents visualized 2D imaging spectra of Li-$K$ (61 eV) and La-$N_{4,5}$ (117 eV) around the S-CE/S-PE interface (Supplementary Fig. 8). Scanning from S-PE to S-CE, Li-$K$ signals were detected throughout the entire region, and La-$N_{4,5}$ signals began to appear from a certain position, which identifies the existence of the S-CE/S-PE interface. A stronger Li-$K$ signal appeared on the S-PE side of

the interface and a weaker signal on the S-CE side, indicating the formation of a space charge region consisting of a Li⁺ accumulation layer and a depletion layer. The Li⁺ redistribution at the interface is caused by the difference in the Li⁺ concentration between the S-CE and S-PE; Li⁺ with a higher concentration in the S-CE ceramic (LLZTO, 23 nm⁻³ for 0.6 mol Ta substitution) diffuses toward the S-PE phase with a lower concentration (1.49 nm⁻³) to compensate for the chemical potential difference, as predicted by space charge models[28,29].

## Properties of S-CE/S-GE interface

We analyzed the coordination structure of the liquid electrolyte components in the two composite layers using Raman spectroscopy. The Raman spectra of the liquid electrolyte-free composite layers (S-CE/B-PE and S-CE/S-PE), the electrolyte-swollen composite layers (S-CE/B-GE and S-CE/S-GE), and the liquid electrolyte (1.5M LiFSI DME) were compared, as shown in Fig. 2a. The signals from FSI⁻ (720 cm⁻¹) and DME solvent (from 830 to 890 cm⁻¹) enable the monitoring of the solvation structure of Li⁺ as FSI⁻ and DME are the components responsible for solvating Li⁺ in the composite layers. The comparison between the S-CE/B-PE and S-CE/B-GE shows that the signal from FSI⁻ appeared after the electrolyte swelling, indicating that the FSI⁻ readily permeates through the S-CE/B-GE. In contrast, the S-CE/S-GE showed a much weaker FSI⁻ signal compared to the liquid electrolyte and S-CE/B-GE. This suggests that the mobile anions (FSI⁻) hardly permeate through the S-CE/S-GE layer owing to the presence of the fixed STFSI⁻ anions. The liquid electrolyte and S-CE/B-GE did not show any notable differences in the Raman signals from DME; the signal from free DME was more intense than that from the coordinated one. Remarkably, the S-CE/S-GE exhibited a stronger signal for free DME and a weaker signal for coordinated DME compared to the liquid electrolyte and S-CE/B-GE. The reduced intensities of FSI⁻ and coordinated DME suggest that the liquid electrolyte molecules are weakly coordinated with Li⁺ in the S-CE/S-GE composite layer.

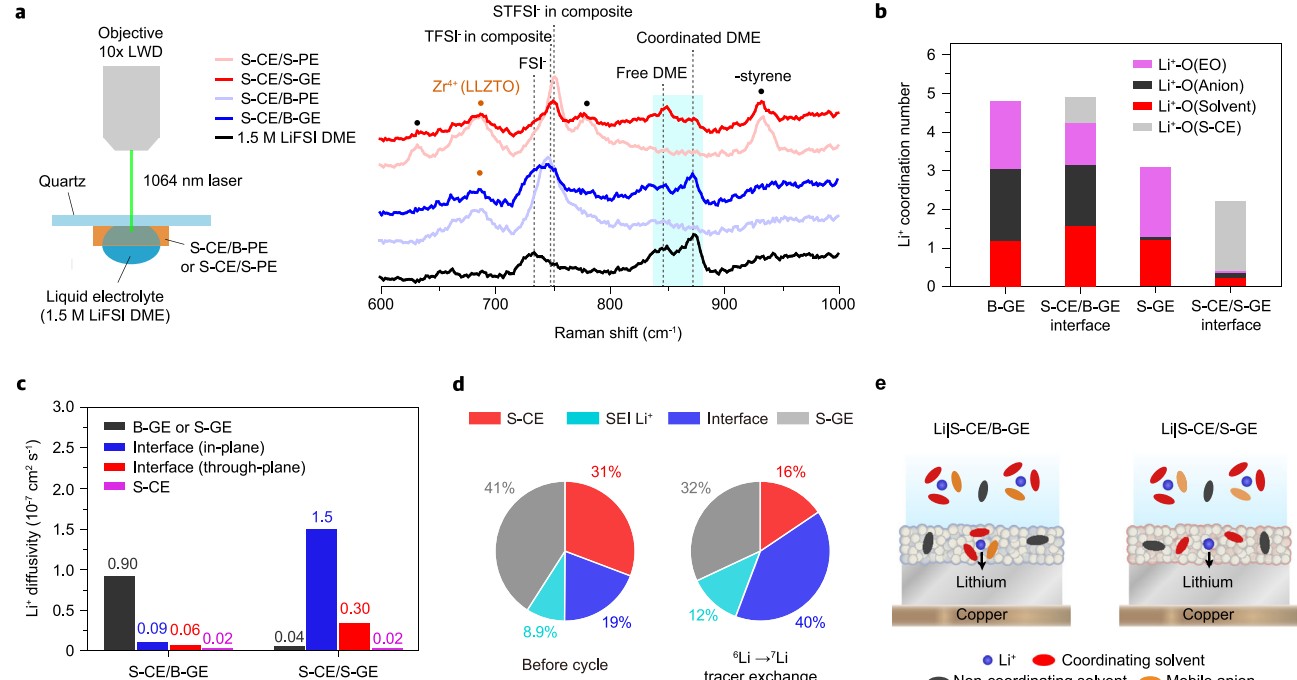

**Fig. 2 | Coordination behaviors of Li⁺ at the S-CE/B-GE and S-CE/S-GE interfaces. a** Raman spectra of the liquid electrolyte (1.5 M LiFSI DME), S-CE/B-PE, S-CE/B-GE, S-CE/S-PE, and S-CE/S-GE. The Raman laser records the ratio of the chemical state (e.g., free DME/coordinated DME or species of anions) of the liquid electrolyte permeated into the composite layer. **b** Contributions to the coordination number of Li⁺ in B-GE, S-CE/B-GE interface, S-GE, and S-CE/S-GE interface. **c** Diffusivities of

Li⁺ in S-CE/B-GE and S-CE/S-GE structures. The diffusivity of Li⁺ at the interface was calculated isotropically in the in-plane and through-plane directions, respectively. **d** ⁶Li MAS-NMR spectra of S-CE/S-GE composite with liquid electrolyte before and after ⁶Li exchange (0.05 mA cm⁻² for 50 h). Pie charts of ⁶Li distribution before and after ⁶Li exchange are presented. **e** Schematic for the coordination state of moving Li⁺ in the composite layer.

An MD simulation was performed to understand the coordination behaviors of electrolyte molecules in the S-CE/B-GE and S-CE/S-GE. The structures of S-CE/B-GE and S-CE/S-GE were constructed such that the Li$^+$ concentration of B-GE and S-GE are identical at 1.5 M (i.e., number density is 0.90 Li$^+$ nm$^{-3}$) in order to circumvent the concentration effect. As shown in Supplementary Fig. 9 and Supplementary Note 1, both interfaces show a Li$^+$ accumulation layer formed at the B-GE or S-GE side in close proximity to the interface[30]. Remarkably, the S-CE/S-GE interface has a higher Li$^+$ number density of 29 Li$^+$ nm$^{-3}$ compared with the S-CE/B-GE interface (6.1 Li$^+$ nm$^{-3}$), emphasizing the critical role of the S-GE in increasing the amount of space charge at the interface. Since the number density of Li$^+$ at the S-CE/S-GE interface is much higher than that of solvents or anions, it can be expected that Li$^+$ at the S-CE/S-GE interface would have a low coordination number with electrolyte molecules. The coordination numbers of Li$^+$ by oxygen atoms of anions, solvent, ethylene oxide group (EO) in polymer, and S-CE were calculated (Fig. 2b and Supplementary Fig. 10). For the B-GE, coordination numbers of oxygen atoms of solvent, anion, and EO are 1.0, 1.9, and 1.4, respectively, and, for the S-CE/B-GE interface, 1.5, 1.6, and 1.1 respectively. The coordination number of S-CE for the S-CE/B-GE interface is merely 0.64, indicating that the interaction between the oxygen of S-CE and Li$^+$ in the accumulation layer is not significant. The S-GE exhibits coordination numbers of solvent, anion and EO of 1.2, 0.08, and 1.8, respectively. Note that at the S-CE/S-GE interface, Li$^+$ has a relatively low coordination number with electrolyte molecules, 0.35, 0.12, and 0.09 for the solvent, anion, and EO, respectively. MD results show that the highly concentrated Li$^+$ space charge layer generated at the S-CE/S-GE interface hinders Li$^+$ coordination by solvent molecules. We are assuming that the coordination structure of Li$^+$ present at the interface between S-CE and S-GE contributed significantly to the Raman signal. The comparison of Li$^+$ solvation free energy further supports the weakly coordinated state of Li$^+$ at the S-CE/S-GE interface (Supplementary Fig. 11 and Supplementary Note 2).

The coordination structure of Li$^+$ present in the dominant transport pathway will have a dominant influence on the anode interfacial properties. In order to confirm the contribution of the Li$^+$ space charge at the S-CE/S-GE interface to the total transport of Li$^+$, we calculated Li$^+$ diffusivities at the S-CE/B-GE and S-CE/S-GE structures (Supplementary Fig. 12). The calculated diffusivities of the bulk phase of the B-GE and S-CE were $0.90 \times 10^{-7}$ cm$^2$ s$^{-1}$ and $0.02 \times 10^{-7}$ cm$^2$ s$^{-1}$, respectively (Fig. 2c), which are similar to the diffusivity of general gel electrolytes ($\sim 10^{-7}$ cm$^2$ s$^{-1}$) or LLZO ($\sim 10^{-9}$ cm$^2$ s$^{-1}$) reported in the literature[31,32], demonstrating the validity of the MD simulation. The Li$^+$ diffusivity in the bulk B-GE is higher than that along or perpendicular to the S-CE/B-GE interface. This result is consistent with previous findings of a small contribution of the S-CE/B-GE interface to Li$^+$ conductivity[33,34]. In sharp contrast, for the S-CE/S-GE interface, the Li$^+$ diffusivity is much higher along the interface ($1.5 \times 10^{-7}$ cm$^2$ s$^{-1}$) than that perpendicular to the interface ($0.30 \times 10^{-7}$ cm$^2$ s$^{-1}$) or in the S-GE phase ($0.04 \times 10^{-7}$ cm$^2$ s$^{-1}$). This indicates that Li$^+$ transport along the S-CE/S-GE interface is dominant in the composite. The ionic conductivities of S-CE, S-GE, and S-CE/S-GE composite in Supplementary Fig. 13 support this calculation. The Li$^+$ conduction pathway of the S-CE/S-GE composite layer was directly scrutinized by tracking $^6$Li$^+$ conduction through the S-CE/S-GE composite from the $^6$Li metal electrode upon stripping (Fig. 2d and Supplementary Fig. 14). As shown in the $^6$Li solid-state NMR spectra for the S-CE/S-GE composite, $^6$Li peaks from S-CE, S-GE, interface, and SEI Li$^+$ were identified, as reported in previous NMR studies[34,35]. The $^6$Li spectrum recorded after the stripping at 2.5 mAh cm$^{-2}$ revealed that the relative peak area for the interface increased from 19 to 40%, in contrast to the decrement of the S-CE peak from 31 to 16% and of the S-GE peak from 41 to 32%. These results support that the S-CE/S-GE interface is a dominant pathway for moving Li$^+$ in the composite layer as calculated in the MD simulation. Raman experiment and MD simulation show the weakly coordinated state of moving Li$^+$ and the free state of electrolyte molecules in the S-CE/S-GE composite layer (Fig. 2e).

## Li anode properties driven by the S-CE/S-GE composite layer

We investigated how the weak coordination of Li$^+$ in the S-CE/S-GE affects the reductive decomposition of the electrolyte on the anode and consequent SEI formation. As illustrated in Fig. 3a, we set up three cells with different electrolyte states (liquid electrolyte, S-CE/B-GE, and S-CE/S-GE) between Li and Cu, and observed the reduction current of electrolyte molecules on the Cu surface. We first performed linear sweep voltammetry (LSV) for Li∥Cu cell to observe the electrolyte reduction behavior. As shown in Fig. 3b, while the S-CE/B-GE shows similar behavior without the composite layer, the introduction of the S-CE/S-GE largely depressed the signals for anion and solvent reduction (highlighted by blue shadow). The lower reduction currents can be understood by diminished anion permeation into the S-CE/S-GE and decreased population of Li$^+$-coordinated DME in the S-CE/S-GE. For further support, the electrolyte reduction current was measured by applying 0 mV to the Li∥Cu cell (i.e., the Cu surface is equipotential with Li). As shown in Fig. 3c, bare Cu (without a composite layer) and the S-CE/B-GE layer showed electrolyte reduction currents of 3.43 µA cm$^{-2}$ and 2.96 µA cm$^{-2}$, respectively, after 40 h, showing no significant difference. This result can be explained by the fact that S-CE/B-GE has a Li$^+$ coordination structure similar to that of a bulk liquid electrolyte according to the MD calculation results. In contrast, the S-CE/S-GE composite layer showed an electrolyte decomposition current of 0.25 µA cm$^{-2}$, which was about an order of magnitude lower than the case without the composite layer or S-CE/B-GE composite layer. The impedance and coulometry results support that the weak coordination of Li$^+$ in the S-CE/S-GE composite layer can alleviate the decomposition of the electrolyte.

The composition of the SEI layer and the morphology of the deposited Li are closely related to the coordination structure of Li$^+$[20–22]. We note that the modulated coordination structure derived by the S-CE/S-GE composite layer enables uniform morphology of the deposited Li. The morphology of the Li deposited under the S-CE/S-GE composite layer, which features compactly packed large Li grains, is much smoother and flatter compared with that on the bare Li electrode and under the S-CE/B-GE (Fig. 3d). The morphological stability of Li was further evaluated through Li symmetric cells with initial Li reservoir of 8 mAh cm$^{-2}$ in each electrode at high capacity (3 mAh cm$^{-2}$) and high current density (5 mA cm$^{-2}$, Fig. 3e). The introduction of the S-CE/S-GE composite layer leads to cycling stability for 400 h without voltage ramping up or short-circuit behavior, while the cell overpotential suddenly increased at around 140 h in the absence of the composite layer. The cell with S-CE/B-GE coated Li failed at 200 h, indicating that the Li$^+$ coordination structure in the composite layer has a clear effect on Li stability.

The residual amount of the main solvent and anions in the Li symmetric cell after cycling was quantified using an NMR analysis (Fig. 3f). For the bare Li cell, ~2.2 mol of solvent was consumed during 150 h cycling, whereas for S-CE/S-GE cell, 1.0 mol of solvent was consumed during 400 h cycling. The anions remained at a level of 46% in the bare Li cell, while 89% remained in the S-CE/S-GE cell. These results indicate that the solvent and anion consumption per cycle is much lower with the S-CE/S-GE composite layer. Note that the proportion of fluorinated solvent among electrolyte molecules consumed during cycling increases from 59% to 79% as the decomposition of anions and solvents decreases with the introduction of the S-CE/S-GE composite layer (inset pie chart). This indicates that the contribution of the fluorinated solvent to SEI formation increases, which can be cross-validated from the chemical composition of the SEI obtained by XPS (Supplementary Figs. 15 and 16). The SEI structure, which is mainly derived from fluorinated solvents, enables the uniform morphology of the deposited Li under the S-CE/S-GE composite layer[23,24,31].

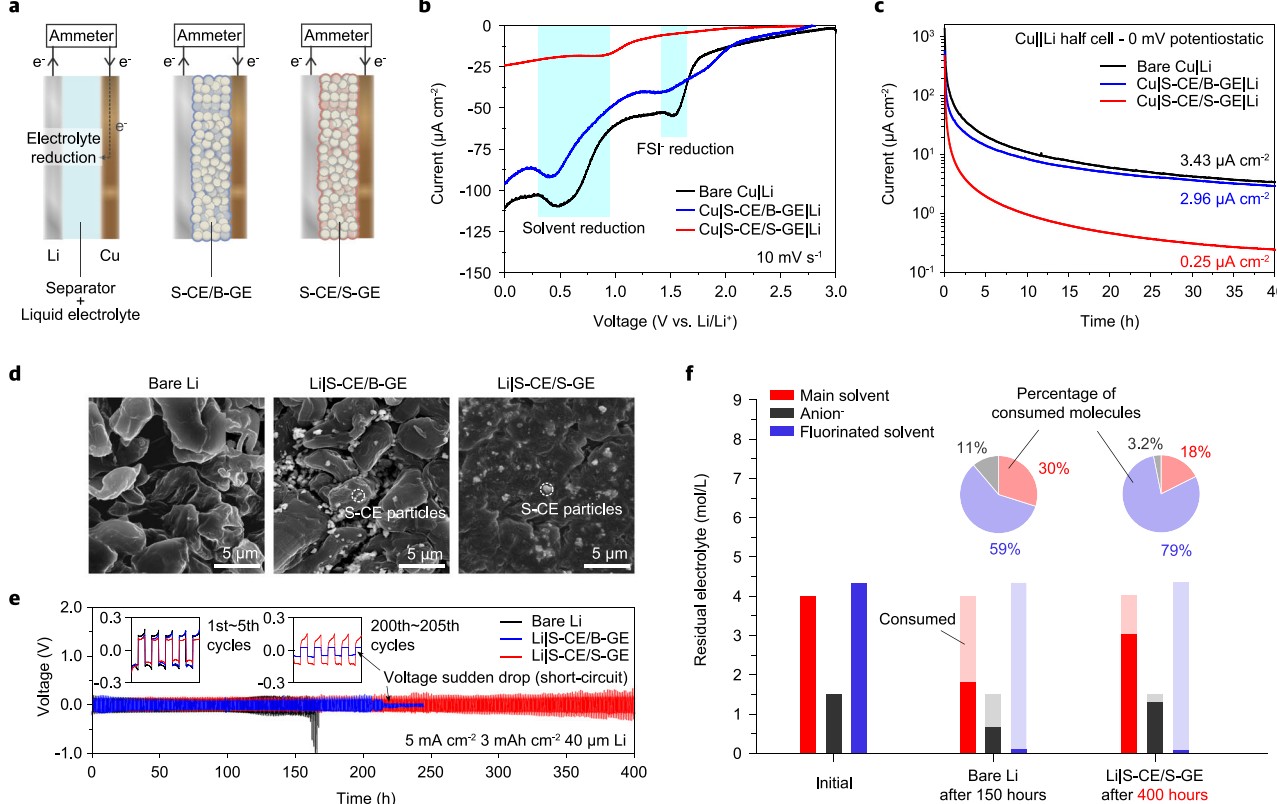

**Fig. 3 | Electrolyte decomposition and reversibility of Li anodes. a** Schematic for measurement of electrolyte reduction current. **b** Measurement of electrolyte reduction current via LSV with Li‖Cu cells. Bare Cu was compared with S-CE/B-GE coated Cu and S-CE/S-GE coated Cu. **c** Measurement of electrolyte reduction current via CA with Li‖Cu cells. Bare Cu was compared with S-CE/B-GE coated Cu and S-CE/S-GE coated Cu. **c** Changes in the coordination state of Li+ in the composite layer and their effects. **d** Deposited Li morphology plated at 3 mAh cm⁻² and 1 mA cm⁻². **e** Bare Li, Li‖S-CE/B-GE, and Li‖S-CE/S-GE symmetric cell at 3 mAh cm⁻² and 5 mA cm⁻². **f** Quantification of the molar concentration of electrolyte remaining in the cell after evaluation of the symmetric cell. The amount consumed is shown translucent. Inset: Pie chart of the percentage of consumed molecules.

## Li‖S-CE/S-GE‖NMC full cell based on designed carbonate electrolyte

To demonstrate the efficacy of the S-CE/S-GE composite layer in the full cell, Li‖NMC532 and Li‖S-CE/S-GE‖NMC532 were tested. We formulated a carbonate electrolyte that enables stable full cell operation, taking into account the role of the S-CE/S-GE. Our formulation utilized a blend of ethylene carbonate and dimethyl carbonate (EC/DMC), which is compatible with the NMC cathode. LiTFSI was adopted, which is relatively reduction-stable with Li metal and chemically stable thanks to the stable −CF₃ group[23,36]. Additionally, we included fluoroethylene carbonate (FEC) as a co-solvent to form an SEI on the Li surface and lithium bis(oxalate)borate (LiBOB) as an additive to form a cathode–electrolyte interphase that prevents oxidative decomposition of the solvent and aluminum corrosion of LiTFSI. The carbonate-based electrolyte also exhibited a weak coordination structure at the S-CE/S-GE interface (Supplementary Figs. 17 and 18), reduced electrolyte decomposition current (Supplementary Figs. 19 and 20), increased contribution of the fluorinated solvent to form SEI (Supplementary Figs. 21 and 22), and improved Li morphology and cycle performance (Supplementary Figs. 23 and 24). The cathode areal capacity, E/C ratio, and N/P ratio of the Li‖NMC and Li‖S-CE/S-GE‖NMC523 cells were 3.63 mAh cm⁻² (4.3 V vs. Li/Li⁺ charge cut-off potential with constant-voltage charging at 4.3 V), 3.5 g Ah⁻¹, and 2.2, respectively, all of which fall within practical levels. The operation of the three replicated Li‖NMC and Li‖S-CE/S-GE‖NMC cells under 0.5 C charge and 1.0 C discharge cycling is shown in Fig. 4a. The capacity of Li‖NMC fully depleted at 50 cycles, which is typical of carbonate electrolytes under a lean electrolyte condition. By contrast, the Li‖S-CE/S-GE‖NMC cell exhibited 91% and 82% capacity retention at 150 and 250 cycles, respectively, under the same conditions. The Li‖S-CE/B-GE‖NMC cell was fully degraded at 60 cycles, indicating that the S-CE/B-GE composite layer does not work strongly in the cell with a low mass of liquid electrolyte content. The full cell cycling stability of the ether electrolyte was also notably enhanced with the S-CE/S-GE (Supplementary Fig. 25).

To observe the suppressed solvent and anion decomposition by the S-CE/S-GE composite layer in the full cell, we traced the residual amount of electrolyte components upon cycling for Li‖NMC532 and Li‖S-CE/S-GE‖NMC532 coin cells by using ¹H, ¹¹B, and ¹⁹F NMR spectroscopy (Fig. 4b). With the S-CE/S-GE composite layer coated Li anode, the consumption rates of EC, DMC, and LiTFSI were remarkably decreased. For additives in Li‖S-CE/S-GE‖NMC532, ~70% of the initial amount of FEC remained after 50 cycles, contrasting with only 5% retention after 50 cycles for Li‖NMC532. Although LiBOB consumes relatively quickly even when the S-CE/S-GE composite layer is introduced on the Li anode because LiBOB can also be consumed at the cathode[37], the residual amount of LiBOB in Li‖S-CE/S-GE‖NMC was about five times higher than that of the Li‖NMC cells because the LiBOB consumption at the Li anode was reduced.

The delay in the depletion of electrolyte components at the Li anode by the S-CE/S-GE composite layer could be cross-validated by analyzing the NMC cathode. We observed higher Coulombic efficiency (average 99.9% at 2nd-15th cycles) in the full cell with the S-CE/S-GE composite layer than in bare Li and Li‖S-CE/B-GE (Fig. 4c). Since the Coulombic efficiency in Li‖NMC corresponds with reversibility at the NMC cathode during cycling (similar to Coulombic efficiency in Li‖

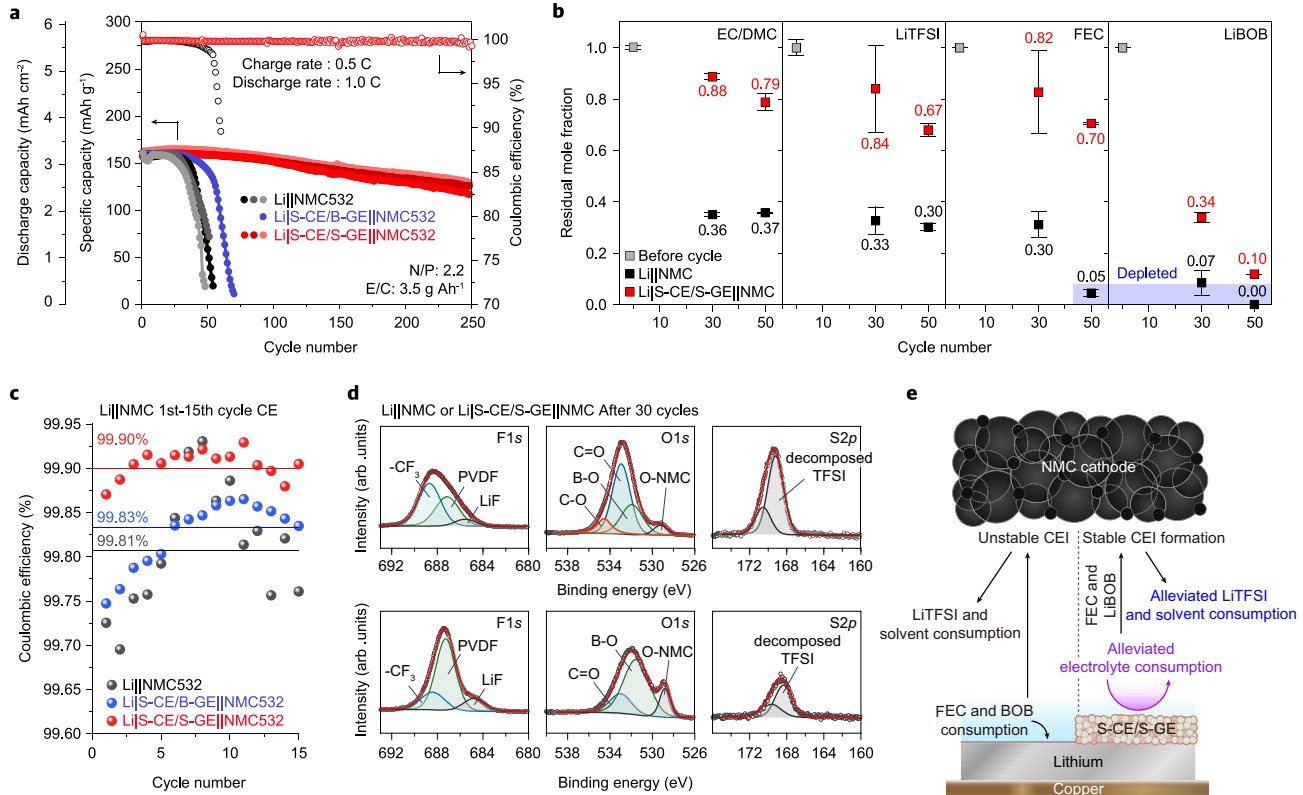

**Fig. 4 | Electrolyte retention and CEI components in Li‖NMC. a** Cycling performance of Li‖NMC532, Li|S-CE/B-GE‖NMC532 and Li|S-CE/S-GE‖NMC532 coin cells having N/P of 2.2 and E/C of 3.5 g Ah⁻¹ measured at 0.5 C constant current and 4.3 V constant voltage (CC-CV) charging and 1.0 C discharging at 25 °C. Reproducibility was confirmed with three replicated cells. **b** Residual mole fractions of EC, DMC, LiTFSI, FEC, and LiBOB remaining in Li‖NMC532 and Li|S-CE/S-GE‖NMC532 coin cells at various cycles. **c** Coulomb efficiency values in the initial cycles (2nd–15th) of

operated full cells. **d** XPS spectra of the NMC532 cathodes for Li‖NMC532 (top) and Li|S-CE/S-GE‖NMC532 (down) cell after 30 cycles. **e** Depiction of the working mechanism of the S-CE/S-GE composite layer in Li‖NMC full cell. The alleviation of the electrolyte decomposition at the Li anode increases the retention of FEC and LiBOB in the cell and allows the formation of a stable CEI, which in turn delays the consumption of LiTFSI and solvent molecules.

graphite cell meaning reversibility of graphite), it could be explained that the increased electrolyte retention by the S-CE/S-GE composite layer affected the cathode stability. We compared the CEI components of Li‖NMC and Li|S-CE/S-GE‖NMC cells after 30 cycles (Fig. 4d). The surface of NMC operated in the Li|S-CE/S-GE‖NMC cell shows higher intensities of materials of the pristine NMC electrode such as PVDF (687 eV) and NMC (529 eV), indicating that a thin CEI was formed without aggressive decomposition of the electrolyte. The weak signals of C-O, C=O (535 eV and 533 eV, decomposition of carbonate solvents), and −CF₃ and the S2p signal (decomposition of LiTFSI) support suppressed electrolyte decomposition at the NMC electrode. In addition, the higher signals of B-O (528 eV, decomposition of LiBOB) and LiF (685 eV, decomposition of FEC) for the Li|S-CE/S-GE‖NMC cell indicate that LiBOB and FEC are more involved in the CEI formation. The Coulombic efficiency and XPS results support that the delayed depletion of FEC and LiBOB by the S-CE/S-GE composite layer on the Li anode enables the formation of a stable CEI at the cathode, which in turn reduces the decomposition of LiTFSI or carbonate solvent (i.e., EC or DMC), as illustrated in Fig. 4e. This virtuous cycle is expected to enable extended cycle life of the Li|S-CE/S-GE‖NMC full cell with low liquid electrolyte content.

### Performance of Li|S-CE/S-GE‖NMC pouch cell

We constructed a pouch-type Li|S-CE/S-GE‖NMC mono-cell based on a carbonate electrolyte to demonstrate the practical applicability of the S-CE/S-GE composite layer (Fig. 5a). To attain intimate contact at the Li and S-CE/S-GE composite layer interface and prevent structural collapse of the S-CE/S-GE composite layer during cell operation[38,39], we

applied a uniaxial pressure of ~280 kPa to the pouch cell using a pressure jig (Supplementary Fig. 26)[5,40]. The areal capacity of the cathode, E/C ratio, and N/P ratio were 3.72 mAh cm⁻², 2.15 g Ah⁻¹, or 1.28 g Ah⁻¹ (1 M LiTFSI EC/EMC + 10% FEC 3% LiBOB, 2.44 or 1.57 g Ah⁻¹ including mass of the composite layer), and 2.15 (40 μm Li; Fig. 5b), respectively, all of which fall within practical levels. At 25 °C, the fabricated cell delivered a specific capacity of 184.3 mAh g⁻¹ at 0.1 C based on the weight of the cathode active material at the first cycle (Fig. 5c). The estimated specific energy density and volumetric energy density of the Li|S-CE/S-GE‖NMC pouch cell were 357 Wh kg⁻¹ and 1041 Wh l⁻¹ at E/C of 2.15 g Ah⁻¹, and 389 Wh kg⁻¹ and 1041 Wh l⁻¹ at E/C of 1.28 g Ah⁻¹, respectively (Fig. 5b and Supplementary Table 1), which are comparable with recently reported state-of-the-art liquid electrolyte-based LMBs[5,19,41–44]. Figure 5c shows the discharge curves of Li|S-CE/S-GE‖NMC532 at various discharging rates from 0.1 C (0.35 mA cm⁻²) to 3.0 C (10.5 mA cm⁻²) at 25 °C in a voltage range of 3.0–4.3 V. The rate capability of Li|S-CE/S-GE‖NMC532 was very close to that of Li‖NMC532 up to 3.0 C rate discharge (Supplementary Fig. 27).

The capacity retention of Li‖NMC, Li|S-CE/B-GE‖NMC, and Li|S-CE/S-GE‖NMC are respectively plotted as a function of the cycle number in Fig. 5d. The Li‖NMC pouch cell exhibited a capacity drop after only 30 cycles with increasing cell overpotential (Fig. 5e). The cycling stability of Li|S-CE/B-GE‖NMC was inferior to that of Li‖NMC under the condition in which the liquid electrolyte content is low, implicating that the liquid electrolyte uptake by the composite layer imposes a burden on the electrode and separator wetting. In sharp contrast, the Li|S-CE/S-GE‖NMC pouch cell showed outstanding cycling stability, as indicated by capacity retention of 97% at 150 cycles

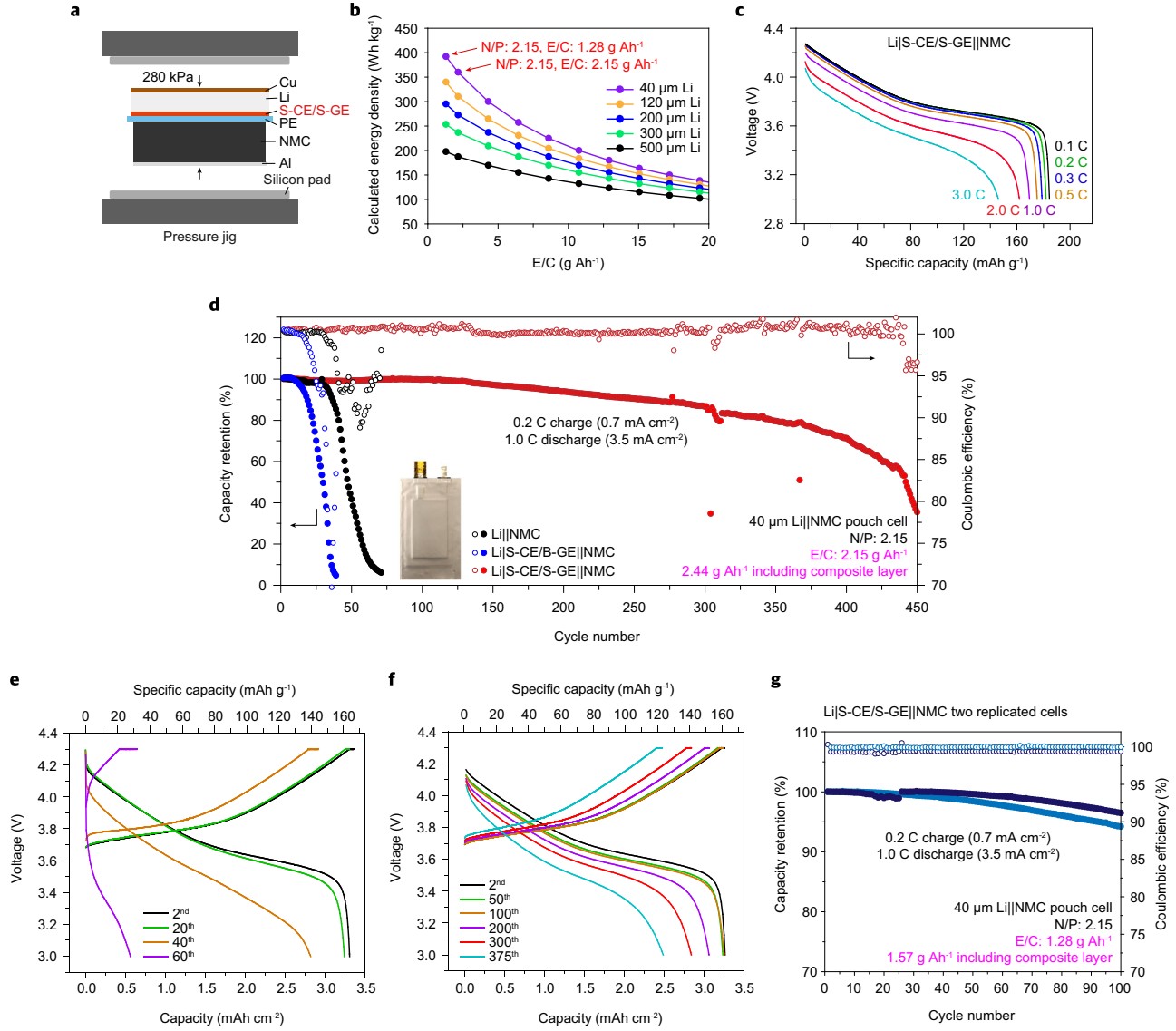

**Fig. 5 | Electrochemical performance of Li||NMC pouch cells. a** Pressurized Li|S-CE/S-GE||NMC pouch cell configuration. **b** Plot of the projected energy density of LMB with E/C ratio at various Li metal electrode thicknesses. **c** Rate capability of Li|S-CE/S-GE||NMC pouch cell at various discharge rates (0.1–3.0 C, 1.0 C = 3.5 mA cm⁻²). **d** Plots of capacity retention as a function of cycle number for the Li||NMC, Li|S-CE/B-GE||NMC and Li|S-CE/S-GE||NMC pouch cells at E/C of 2.15 g Ah⁻¹ (2.44 g Ah⁻¹ including composite layer) operated at a charge/discharge rate of 0.2 C/1.0 C in a voltage window of 3.0–4.3 V with constant voltage charging at 4.3 V at 25 °C. **e, f** Voltage profiles of the Li||NMC and Li|S-CE/S-GE||NMC pouch cells at selected cycles. **g** Plots of capacity retention as a function of cycle number for the Li|S-CE/S-GE||NMC pouch cells at E/C of 1.28 g Ah⁻¹ (1.57 g Ah⁻¹ including composite layer) operated at a charge/discharge rate of 0.2 C/1.0 C in a voltage window of 3.0–4.3 V with constant voltage charging at 4.3 V at 25 °C. The results of two samples are shown.

and 80% at 350 cycles. These results indicate that the electrolyte depletion can be retarded in the Li|S-CE/S-GE||NMC pouch cell even with the low liquid electrolyte content (E/C of 2.15 g Ah⁻¹). The voltage profile of the Li|S-CE/S-GE||NMC cell remained stable over 350 cycles (Fig. 5f), indicating that the thickness of the Li electrode increases slowly (Supplementary Fig. 28) and no exhaustion of the Li electrode or electrolyte occurred. In addition, Li|S-CE/S-GE||NMC cells outperformed Li||NMC cells injected with excess electrolyte (Supplementary Fig. 29). Remarkably, the Li|S-CE/S-GE||NMC cell stably operated up to 100 cycles even lower liquid electrolyte content (E/C of 1.28 g Ah⁻¹, Fig. 5g), demonstrating the effectiveness of the S-CE/S-GE composite layer in achieving energy-dense LMBs. The remarkable cycling stability of the full cell with the S-CE/S-GE composite layer, despite the E/C ratio of this work being the lowest value for LMBs with carbonate electrolyte[5,19,21,41,43–49], underscores the practical advantages of the S-CE/S-GE composite layer.

## Discussion

In this work, the efficacy of an S-CE/S-GE composite layer as a protective layer in enhancing the cycling stability of LMBs was demonstrated, and its underlying mechanism was elucidated. We unraveled that, in the composite, Li⁺ is in a weakly coordinated state and the alleviated coordination between Li⁺ and electrolyte molecules significantly reduces the irreversible decomposition of the liquid electrolyte on the anode. The weak coordination of the Li⁺ enables a dense, dendrite-free morphology of deposited Li with minimal electrolyte consumption. A pouch cell of Li|S-CE/S-GE||NMC exhibited stable operation under low liquid electrolyte content. Our work highlights the importance of reducing electrolyte consumption while inducing uniform plating of Li for the long-term operation of cells with low liquid electrolyte content and the impact of modulating the Li⁺ coordination structure at the Li electrode surface for this purpose.

Based on the weakly coordinated Li$^+$ structure due to the interplay between the liquid electrolyte and composite single-ion-conductor examined in this study, we expect that the efficacy of the composite single-ion conductor can be amplified by designing liquid electrolyte.

## Methods

### Preparation of the S-CE/S-PE composite layer

The S-CE/S-PE composite layer was fabricated via the following procedure. $Li_{6.4}La_3Zr_{1.4}Ta_{0.6}O_{12}$ NPs (S-CE, 500 nm, Ampcera$^{TM}$) and (STFSI)Li monomer (synthesized by KRICT, Korea) were mixed in EC/DEC (Sigma-Aldrich) and constantly stirred in an Ar-filled glove box. PEGDA-575 ($M_n$: 575, Sigma-Aldrich) in an EO:Li molar ratio with S(TFSI)Li as about 1:1 mol% was added to the mixture under stirring to prepare a dispersion slurry. The weight ratio of LLZTO:(STFSI)Li:PEGDA in the slurry is 4.0:0.9:0.1. The solid content of the S-CE/S-PE slurry is about 50 weight ratio. The slurry was then cast on a Li foil using a doctor blade. The casting thickness was 20 μm. The cast on the Li foil was heated at 60 °C in an Ar-filled glovebox for 10 h to evaporate the casting solvent. The thickness of the layer was 6–7 μm. The S-CE/B-PE composite layer consisting of LLZTO, LiTFSI, and PEGDA was manufactured through the same process as employed for the S-CE/S-GE composite layer. The weight fraction of the LLZTO, LiTFSI, and PEGDA in EC/DEC for the S-CE/B-PE composite layer is equal to that for the S-CE/S-PE composite layer. All fabrication processes were conducted in an Ar-filled glovebox.

### Li and S-CE/S-PE composite layer morphology characterization

A SEM analysis was performed using a Field-Emission Scanning Electron Microscope (FE-SEM, Sirion by FEI). The top view of the Li deposition morphologies could be observed due to the transfer of some parts of the composite layer to the separator. The cross-sectional view of the S-CE/S-PE composite layer was observed via a focused ion beam system (FIB-SEM, Helios Nanolab 450 F1, FEI), and surface cleaning was performed at 5 kV to remove a layer damaged by Ga ions.

### TEM, EDX, and SR-EELS imaging

TEM and STEM-EDX analyses were performed using a Transmission Electron Microscope (TEM, Tecnai G2 F30) with 200 kV accelerating voltage. Spatially resolved electron energy-loss spectroscopy (SR-EELS) was performed by an Ultra-Corrected-Energy-Filtered Transmission Electron Microscope (UC-EF-TEM, Libra 200 HT Mc Cs) with exposure time of 0.01 s to obtain EELS spectra. The intensity of each element was modified to represent the number density ($N$) according to the Eq. (1).

$$N \propto \frac{I_k}{I_{total}} \tag{1}$$

Where $N$ represents the number density of atom, $I_K$ represents the integral of edge count of the shell (Li-$K$ or La-$N_{4,5}$), and $I_{total}$ represents the integral of total counts in the spectrum. TEM samples were prepared as follows. The S-CE/S-PE composite was ground and dispersed in EC/DEC solvent. The solution was sonicated for 5 min and then loaded on the TEM grid. Because the S-CE particles were uniformly coated by S-PE, the brief exposure of the sample to air during TEM grid loading did not produce any byproducts.

### NMR experiments

$^1H$, $^{11}B$, and $^{19}F$ NMR were performed to measure the remaining electrolyte molecules in cycled cells using a Liquid 400 MHz Nuclear Magnetic Resonance spectrometer (Liquid 400 MHz NMR. Agilent 400 MHz 54 mm NMR DD2). After washing the cycled coin cell with 1.5 ml of 0.1 M fluorobenzene and 0.1 M LiBF$_4$ in DMSO-d6, the washing solution (i.e., DMSO-d6 solution with residual electrolyte) was gathered for the NMR analysis. The residual amount of each component can be quantified by calculating the relative intensities with respect to an internal reference; dimethyl sulfoxide-d6 for the main solvent (DME, EC, or DMC, in $^1H$

spectra), fluorobenzene for fluorinated solvents (TFTFE or FEC, in $^{19}F$ spectra) and anion (LiFSI or LiTFSI in $^{19}F$ spectra), and LiBF$_4$ for LiBOB (in $^{11}B$ spectra). The residual mole fraction was obtained by normalizing the amount of each electrolyte component to the amount before cycling. The solid-state $^6Li$ magic-angle-spinning (MAS) NMR measurement was conducted under 25 kHz MAS on a 400 MHz Solid-State NMR spectrometer (AVANCE III HD, Bruker) at the Korea Basic Science Institute (KBSI) Western Seoul Center. For the MAS NMR measurement, the pristine S-CE/S-GE composite (1.5 M LiFSI DME/TFTFE was used as a liquid electrolyte) and that from the operated cell were dried to remove DME and TFTFE. $^6LiFSI$ in the LiFSI precipitate formed during drying represents Li-ion conduction through the S-GE phase and liquid phase. The S-CE/S-GE samples were then ground into small particles using a mortar. The samples were subsequently put into a 1.9 mm zirconia rotor and NMR analyses were performed.

### Raman spectroscopy

The S-CE/B-PE and S-CE/S-PE solid composite for Raman analysis were fabricated by disposing of slurry on the bottom part of the coin cell and drying for 48 h. The dried composites were attached to the quartz cover glass, and wetted in a liquid electrolyte (1.5 M LiFSI DME) to form an S-CE/B-GE or S-CE/S-GE state as shown in Fig. 2a. Raman spectra of the prepared samples were obtained through Raman spectroscopy (LabRAM HR Evolution Visible_NIR, HORIBA). A laser of 1064 nm wavelength was used, and the signal was integrated for 60 s to obtain Raman spectra.

### Pouch cell preparation

A prototype pouch cell was fabricated using Li| S-CE/S-GE. The Li| S-CE/S-GE anode and NMC cathode were punched into 40 mm × 60 mm and 30 mm x 50 mm-sized electrodes, respectively. The Cu foil and NMC cathode were welded with Ni and Al tabs by an ultrasonic welder. All the electrodes and separators were stacked and packed in an aluminum pouch bag. Next, 100 μl (2.15 g Ah$^{-1}$) or 60 μl (1.28 g Ah$^{-1}$) of 1 M LiTFSI EC/DMC + 10% FEC + 3% LiBOB electrolyte was injected into the cell before sealing the pouch. After 1 day of aging, the pouch cells were subjected to degassing and vacuum resealing. All the cell assembly processes were conducted in an Ar-filled glovebox. The fabricated LMB pouch cells were uniformly pressurized by a pressure jig with four clamping bolts and nuts and a pressure distribution of a 1-mm-thick silicon pad. The clamping torque was 50 kgf·cm. A WBCS3000L battery tester was used for the cell tests under 25 °C.

### Electrochemical test

The cells for ion conductivity, solid-state $^6Li$ MAS NMR measurements, LSV, and CA experiments were fabricated using the CR 2032-type coin cell format. A SUS blocking electrode (ionic conductivity), Li electrode (NMR), or Cu electrode (LSV and CA) was placed on the bottom of the coin cell, and S-CE/S-PE slurry was disposed on it. After drying the slurry, a predetermined amount of liquid electrolyte (the weight ratio of liquid electrolyte to S-PE is 1.59 for 1.5 M LiFSI DME +50% TFTFE, and 2.37 for 1 M LiTFSI EC/DMC + 10% FEC + 3% LiBOB) was added to the cell to form S-CE/S-GE. The ratio was determined based on the equilibrium swelling of the S-PE in the liquid electrolyte. A SUS electrode (ionic conductivity), $^6Li$ electrode (NMR) or Li electrode (for LSV and CA) was then placed on the top, and the cell was clamped. The transference number was also measured in the same way, but in the solid-state state (i.e., S-CE/B-PE and S-CE/S-PE) to measure the ion conduction properties of the composite itself. The symmetric cell was assembled with two pieces of 40 μm Li foil or composite layer coated Li anodes, 100 μl of 1.5 M LiFSI DME + 50% TFTFE, and a 19pi-12pi O-ring. A coin-type full cell (Li-NMC) test was conducted with a 2032 coin-type cell using $LiNi_{0.5}Mn_{0.3}Co_{0.2}O_2$ (NMC532) cathode (21.47 mg cm$^{-2}$ areal loading, NMC532:Super C65:PVDF = 94:3:3), Li anode, and PE separator (Asahi, 19 μm). Li||NMC cells were assembled with 1 M LiTFSI EC/DMC + 10% FEC + 3% LiBOB

electrolyte in a voltage range of 3.0–4.3 V at 0.5 C charge and 1 C discharge for coin cells and 0.2 C charge and 1 C discharge for the pouch cells. After the constant current charging step, the constant voltage at 4.3 V was set for potentiostatic charging until the current density reached 0.05 C. All cell assembly processes were conducted in an Ar-filled glovebox. A WBCS3000L battery tester was used for all electrochemical tests under 25 °C unless specified. The impedance measurements were conducted using a Solartron 1470E Frequency Response Analyzer (Solartron Analytical) in a frequency range from 1 MHz to 1 Hz, with a perturbation degree of 10 mV.

### Simulations

All density functional theory (DFT) calculations were performed with the program package DMol[3] in Materials Studio (Accerlrys Inc.). Dmol[3] uses numerical orbitals as basis functions, each of which corresponds to atomic orbitals. These works utilized a double-numeric-plus-polarization (DNP) function and a global orbital cutoff of 4 Å. The size of the DNP basis set is comparable to Gaussian 6-31G(d), but the DNP is more accurate than the corresponding Gaussian basis set. DFT calculations were carried out with a gradient-corrected (GGA) functional with the Perdew-Bueke-Ernzerhof (PBE) exchange correlation functional. Tolerances of energy, gradient, and displacement convergence were 0.00001 hartree, 0.002 hartree/Å, and 0.005 Å, respectively. The force tolerance of self-consistent-field (SCF) cycles was $1.0 \times 10^{-6}$. The Grimme's DFT-D2 method was adopted to account for the Van der Waals interactions (vdW), and this method was optimized for several DFT functionals. All molecular dynamics calculations were performed with the program package Forcite in Materials Studio (Accerlrys Inc.). We set the LLZO slap as the (110) LLZO Li-La layer, which is the most stable cleaved surface, as reported in a previous paper[50]. All components were configured in a simulation cell with geometric optimization and charge applied through DFT. MD was performed using the Forcite module and Forcefield 'COMPASS III'. In the MD dynamics simulation, 5-mer of P(STFSI)Li-PEGDA ($n = 1$) was used as a S-PE for simplicity in calculation (EO:Li=3:1). MD simulations of the LLZO/LiFSI/DME/PEGDA (S-CE/B-GE) and LLZO/P(STFSI)Li-co-PEGDA/DME (S-CE/S-GE) were performed at 298 K and 1 atm. The content of LiFSI in DME was set to a concentration of 1.5 M. For P(STFSI)Li/DME, additional DME molecules were added to the SIP model to make a 1.5 M concentration. The simulation boxes were $63.3 \times 45.3 \times 45.5$ Å[3] for S-CE/B-GE and $63.3 \times 45.3 \times 51.6$ Å[3] for S-CE/S-GE. The O, Zr, and La atoms in LLZO (S-CE) were constrained for faster calculation, considering their extremely low mobility in the LLZO lattice. The simulation boxes were periodic along the x, y, and z directions. The systems were geometrically stabilized using Smart Algorithm employing a convergence tolerance of 0.001 kcal mol$^{-1}$ Å$^{-1}$. They were then equilibrated in NVT ensembles using Nose Algorithm with a Q ratio of 0.1. Before the dynamics calculation, a 1 ns equilibration process was conducted. After equilibrium, a 3 ns NVT run with a 3 fs time step was performed for each interface with sampling in a time interval of 45 fs to collect the simulation data. The radial distribution function (RDF) of the bulk phase and the interphase were calculated by using Eqs. (2) and (3), respectively.

$$g_{bulk\ phase}(\text{r}) = \frac{dn_r}{4\pi r^2 dr * \rho} \quad (2)$$

$$g_{inter\ phase}(\text{r}) = \frac{dn_r}{2\pi r^2 dr * \rho} \quad (3)$$

The coordination number N(r) was calculated by Eq. (4):

$$N(r) = 4\pi\rho \int_0^{r'} g(r)r^2 dr \quad (4)$$

The mean square displacement (MSD) evolution with time was calculated using the built-in analyzer of Forcite in Materials Studio. The MSD of N number of atoms (molecules) in the ensemble for each spatial direction was calculated with Eq. (5).

$$MSD_i = |x_i(t) - x_i(0)|^2,\ i = x, y, z \quad (5)$$

## Data availability

All relevant data that support the findings of this study are presented in the article, Supplementary Information. Source data are provided as a Source data file. Source data are provided with this paper.

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

## Acknowledgements

This work was supported by the KAIST Institute for Nano-Century (KINC), and by the Technology Development Program to Solve Climate Changes through the National Research Foundation of Korea (NRF), which is funded by the Ministry of Science, ICT grants 2018M1A2A2063807 (H.-T.K.).

## Author contributions

H.K., Y.S.M., and H.-T.K. conceived the concept of single-ion-conducting composite layer and designed this work; H.K. carried out the experimental planning, electrochemical measurements, characterization, and data analysis; H.-J.C. assisted fabricated of lithium metal pouch cell; J.J. advised MD and DFT calculations; J.-K.J. synthesized and provided monomer of single-ion-conducting polymer; W.L. and N.-S.C. assisted electrolyte design; J.H., Y.R., J.B., D.J.S., and J.-H.L. contributed to experimental design and discussion of the results; H.K., Y.S.M., and H.-T.K. wrote the manuscript, and H.-T.K. supervised this work; all authors commented on the manuscript.

## Competing interests

The authors declare no competing interests.
