## [Peer Review File · Nature Communications]

Reviewer comments, first round review:

Reviewer #1 (Remarks to the Author):

The manuscript describes an inorganic/gel polymer composite electrolyte that improves Li cycling by suppressing electrolyte consumption. The performance improvement in carbonate-based electrolyte is quite impressive. Analysis of the anode and cathode surfaces indicates that the coating allows the fluorinated component in the electrolyte to play a bigger role in forming CEI and SEI layers. The results will be of interest to the community and the technological advancement is meaningful. The proposed mechanism, on the other hand, is less than fully supported. I would recommend acceptance after a few issues are addressed:

- 1) Since this is a gel electrolyte, I would avoid calling this a single ion coating since it is not.
- 2) I am skeptical of the claim that the weakly solvating structure is responsible for the reduced electrolyte consumption. This has largely been done through a computational study. Since the authors have made bulk quantities of this composite structure, would it be possible to use Raman or other techniques to directly confirm this?
- 3) Do the authors have the cross-sectional images of the cycled Li? Reduction in electrolyte consumption is usually associated with a slower increase in Li thickness.
- 4) What is the equilibrium swelling ratio of the PE? What would happen if there is no CE? Would the mechanism change?

Reviewer #2 (Remarks to the Author):

This manuscript presented a single-ion conduction protective layer for a "lean" electrolyte Li battery. In both the manuscript and the previous response to reviewers' comments, the authors keep emphasizing that the novelty of this piece of work is the prepared LLZO/Polymer layer alleviating the reactivity of liquid electrolyte with Li anode by space charge layer/weakly coordinated Li ions in the "protective layer," therefore enables stable cycling of LEAN electrolyte Li batteries. Although massive amounts of data are provided, the results do not sufficiently support the proposed theory. The cell design repeats previous publications since the decade on quasi-solid Li batteries in which a tiny amount of liquid electrolyte is used to wet the interfaces between electrodes and solid electrolyte. The difference in this draft is that the authors name the prepared (quasi)solid electrolyte layer (S-CE/S-GE) as a "protective layer" on the lithium surface, and the full cell then is LEAN electrolyte due to the small portion of liquid electrolyte. This is a misusage of the "lean electrolyte" concept. Otherwise, is this mean in an all-solid-state battery with only a dense LLZO layer to separate the cathode and metallic lithium anode, one can call it an electrolyte-free battery? In addition, as the previous reviewers mentioned, the presented battery performance is not outstanding. The Li battery with the complicated processing of a "protective layer" is not performing better than that of even using simpler coating compounds like LiF.

The authors emphasize the less liquid electrolyte consumption in the cells with S-CE/S-GE layer is due to weakly coordinated Li ions in the "protective layer". However, the experimental process was suggested differently. As Li metal chemically reacts with the solvents in liquid electrolytes (e.g., with EC <https://pubs.acs.org/doi/10.1021/acsaem.8b01707>), SEI layer will form and consume some electrolyte. When the lithium anode is coated with the S-CE/S-GE layer, a large amount of EC/DEC was introduced on the surface of lithium by tape casting. A layer of reaction products is already formed. This layer protects the lithium when it is in contact with liquid electrolytes. The manuscript has no result or discussion on such chemical aging influence. If the authors think the solvents used to tape cast the S-CE/S-GE layer are stable with metallic lithium, why not directly use a liquid electrolyte with such solvents in full cells? Besides, after coating the "protective layer", the "free volume" of Li in contact with the liquid electrolyte in a battery is much smaller than bare lithium; of course, the consumption of liquid electrolyte is smaller in that case. Taking these points into account, the provided data barely support the mechanism proposed by the authors. The major chemical and physical influences are missing.

Based on the adopted concepts and lacking novelty, considering the significance of this topic and the quality of evidence provided to explain the obtained phenomenon, I suggest rejecting the paper.

Revisions need to be made before submitting this paper to any other journals, including but not limited to the following aspects:

1. In figure 1b, the SEM image doesn't indicate the total thickness of SCL is 6-7 μ m. The edge of the SCL is invisible.
2. The successful synthesis of S-PE is only proved by the IR spectrum in the supporting information. This is not enough since the CH₂ asym stretch is similar in all three samples. Besides, the IR result in red is clearly treated by baseline correction and so on, whereas the results of STFSILi and PSTFSILi were not. How can one compare the results with different treatments?
3. The liquid electrolyte used in symmetric cells differs from the one used in the full cell. How to correlate the characterization results done in symmetric cells to the full cell performance since they are different systems?
4. Fig. 3e. The applied scale is so large that nothing is visible. E.g., what is the voltage difference between the cells? The cells should have different resistance. Why do they seem at the same potential? Besides, what is the reason for the potential increase for the cell with bare Li after 300 hours of cycling?
5. The calculation and comparison with the literature shown in Fig. 5g and Table S2 seem unfair since the "protective layer" is excluded. Besides, the comparison is misleading for pouch cells in Table S2 due to the different cell assembly, cycling protocol and other unspecified vital factors.

Reviewer #1 (Remarks to the Author):

The manuscript describes an inorganic/gel polymer composite electrolyte that improves Li cycling by suppressing electrolyte consumption. The performance improvement in carbonate-based electrolyte is quite impressive. Analysis of the anode and cathode surfaces indicates that the coating allows the fluorinated component in the electrolyte to play a bigger role in forming CEI and SEI layers. The results will be of interest to the community and the technological advancement is meaningful. The proposed mechanism, on the other hand, is less than fully supported. I would recommend acceptance after a few issues are addressed:

RESPONSE: The authors greatly appreciate Reviewer #1 for the detailed feedback and positive recommendation. Reviewer #1 highlighted the need for more evidence to support our proposed mechanism and suggested conducting Raman analysis. In response to this important comment, we performed Raman analysis to confirm the weakly coordinated state in the composite layer. We hope that this addresses the reviewer's concern about the validity of the mechanism.

1) Since this is a gel electrolyte, I would avoid calling this a single ion coating since it is not.

RESPONSE: The authors appreciate the reviewer's concern about the possibility of readers misinterpreting our composite layer as a pure single-ion-conductor (i.e., $t_{Li^+} = 1$). As the reviewer correctly noted, it is physically impossible to prevent any anion permeation into the composite layer.

Please allow us to explain our rationale for the previous naming. The main point of this work is to demonstrate how the coordination structure of the liquid electrolyte changes depending on whether the base material of the composite layer is a single-ion-conductor or a bi-ion-conductor. Therefore, we thought it would be logical to name the composite layer based on its material. However, as the reviewer pointed out, the term 'single-ion-conductive composite layer' that we used before could imply that the gel layer only transports lithium ions. To avoid any confusion, we have replaced the term 'single-ion-conducting composite layer' or similar expressions with 'single-ion-conductor-based composite layer' when referring to a gel-state composite.

2) I am skeptical of the claim that the weakly solvating structure is responsible for the reduced electrolyte consumption. This has largely been done through a computational study. Since the

authors have made bulk quantities of this composite structure, would it be possible to use Raman or other techniques to directly confirm this?

RESPONSE: The authors appreciate the reviewer for initiating a discussion on the correlation between the Li^+ coordination structure and electrolyte consumption. The authors believe that the following discussion has strengthened the mechanism.

In the previous round of the revision, we compared S-CE/B-GE and S-CE/S-GE to highlight the importance of the Li^+ coordination structure to the cathodic decomposition of liquid electrolyte components. The reduced electrolyte consumption for S-CE/S-GE cannot be simply explained by reduced contact between the liquid electrolyte and the Li metal electrode because the S-CE/B-GE also suppresses electrolyte swelling owing to the crosslinked structure of B-GE. Based on the MD simulation, we suggest that the reduced electrolyte consumption observed for S-CE/S-GE is responsible for the weakly solvating structure.

In order to solidify our argument, we analyzed the coordination structure of the liquid electrolyte components in the two composite layers using Raman spectroscopy. The Raman spectra of the liquid-electrolyte-free composite layers (S-CE/B-PE and S-CE/S-PE), the electrolyte-swollen composite layers (S-CE/B-GE and S-CE/S-GE), and the liquid electrolyte (1.5M LiFSI DME) were compared, as shown in **Fig. R1**. The signals from FSI^- (720 cm^{-1}) and DME solvent (from 830 to 890 cm^{-1}) enable the monitoring of the solvation structure of Li^+ as FSI^- and DME are the components responsible for solvating Li^+ in the composite layers.

The comparison between the S-CE/B-PE and S-CE/B-GE shows that the signal from FSI^- appeared after the electrolyte swelling, indicating that the FSI^- readily permeates through the S-CE/B-GE. In contrast, the S-CE/S-GE showed a much weaker FSI^- signal compared to the liquid electrolyte and S-CE/B-GE. This suggests that the mobile anions (FSI^-) hardly permeate through the S-CE/S-GE layer owing to the presence of the fixed STFSI⁻ anions in the layer. The liquid electrolyte and S-CE/B-GE did not show any notable differences in the Raman signals from DME; the signal from free DME was more intense than that from coordinated DME. Remarkably, the S-CE/S-GE exhibited a stronger signal for free DME and a weaker signal for coordinated DME compared to those for the liquid electrolyte and S-CE/B-GE. The reduced intensities of FSI^- and coordinated DME suggest that the liquid electrolyte molecules are weakly coordinated with Li^+ in the S-CE/S-GE composite layer.

We strongly believe that this Raman analysis provides solid evidence for the change in the solvation structure of Li^+ . Therefore, we added it to the revised manuscript, as shown in **Fig. 2a**.

Fig. R1: Coordination structure of the liquid electrolyte components in the S-CE/B-GE and S-CE/S-GE. Raman spectra of the liquid electrolyte (1.5 M LiFSI DME), S-CE/B-PE, S-CE/B-GE, S-CE/S-PE, and S-CE/S-GE.

(Lines 150–166)

We analyzed the coordination structure of the liquid electrolyte components in the two composite layers using Raman spectroscopy. The Raman spectra of the liquid electrolyte-free composite layers (S-CE/B-PE and S-CE/S-PE), the electrolyte-swollen composite layers (S-CE/B-GE and S-CE/S-GE), and the liquid electrolyte (1.5M LiFSI DME) were compared, as shown in **Fig. 2a**. The signals from FSI^- (720 cm^{-1}) and DME solvent (from 830 to 890 cm^{-1}) enable the monitoring of the solvation structure of Li^+ as FSI^- and DME are the components responsible for solvating Li^+ in the composite layers. The comparison between the S-CE/B-PE and S-CE/B-GE shows that the signal from FSI^- appeared after the electrolyte swelling, indicating that the FSI^- readily permeates through the S-CE/B-GE. In contrast, the S-CE/S-GE showed a much weaker FSI^- signal compared to the liquid electrolyte and S-CE/B-GE. This suggests that the mobile anions (FSI^-) hardly permeate through the S-CE/S-GE layer owing to the presence of the fixed STFSI^- anions. The liquid electrolyte and S-CE/B-GE did not show any notable differences in the Raman signals from DME; the signal from free DME was more intense than that from coordinated DME. Remarkably, the S-CE/S-GE exhibited a stronger signal for free DME and a weaker signal for coordinated DME compared to the liquid electrolyte and S-CE/B-GE. The reduced intensities of FSI^- and coordinated DME suggest that

the liquid electrolyte molecules are weakly coordinated with Li^+ in the S-CE/S-GE composite layer.

3) Do the authors have the cross-sectional images of the cycled Li? Reduction in electrolyte consumption is usually associated with a slower increase in Li thickness.

RESPONSE: The authors appreciate the valuable suggestion provided by the reviewer. We operated $\text{Li}||\text{NCM}$ and $\text{Li}|\text{S-CE/S-GE}||\text{NCM}$ pouch cells for 50 cycles under the same conditions employed for the cycling stability test, and then subjected the cycled Li metal electrode to cross-sectional SEM analysis. For the bare Li electrode, the thickness increased from 40.0 to 59.4 μm ($\Delta = 19.4 \mu\text{m}$) after 50 cycles as a result of the formation of a porous Li layer. However, for the $\text{Li}|\text{S-CE/S-GE}$ electrode, the porous Li layer was hardly identified, and the total thickness of the S-CE/S-GE layer-coated Li metal electrode increased from the 46.0 to 52.2 μm ($\Delta = 6.2 \mu\text{m}$). This result indicates that the increase in the thickness of the Li metal electrode was much smaller for the S-CE/S-GE layer, providing further evidence for the suppressed electrolyte decomposition. To investigate the structure of the S-CE/S-GE layer after 50 cycles, we used FIB-SEM and EDS analyses. The uniform La, O, and F distribution and the uniform thickness suggest the stability of the S-CE/S-GE. However, the layer was expanded from 6 to 15 μm thick, probably due to the inclusion of the porous Li components. We added the cross-sectional images to the revised manuscript (**Supplementary Fig. 28**).

Fig. R2. Cross-sectional morphology of the cycled bare Li and $\text{Li}|\text{S-CE/S-GE}$. **a**, Cross-sectional SEM images. Scale bar: 50 μm . **b**, FIB-SEM (52° tilt) and EDS images for the cycled $\text{Li}|\text{S-CE/S-GE}$. Scale bar: 10 μm .

4) What is the equilibrium swelling ratio of the PE? What would happen if there is no CE? Would the mechanism change?

RESPONSE: The authors thank the reviewer for this query. The equilibrium swelling ratio of S-PE is 159% in the ether electrolyte and 237% in the carbonate electrolyte (**Fig. R3**). The equilibrium swelling ratios have been added in the revised manuscript.

For the S-GE layer, which does not contain CE, we observed a significantly larger interfacial resistance (**Fig. R4**). Additionally, we investigated the cycling stability of a Li|Li symmetric cell for a composite layer consisting of Al₂O₃ (480 nm in diameter) and S-GE, and found that the Al₂O₃/S-GE-coated Li exhibited even worse cycling stability compared to the bare Li electrode (**Fig. R5**). These results provide further evidence for the critical role of the S-CE in the underlying mechanism.

Fig. R3. Measurement of the equilibrium swelling ratio for S-PE.

Fig. R4. Nyquist plots for Li|S-GE||S-GE|Li and Li|S-CE/S-GE||S-CE/S-GE|Li.

Fig. R5. Symmetric cell performance for Bare Li, Li|Al₂O₃/S-GE, and Li|S-CE/S-GE at 3 mAh cm⁻² and 5 mA cm⁻². Electrolyte: 1.5 M LiFSI DME + 50% TTFTE.

The authors greatly appreciate the reviewer's feedback, which has improved the quality of our work.

Reviewer #2 (Remarks to the Author):

This manuscript presented a single-ion conduction protective layer for a “lean” electrolyte Li battery. In both the manuscript and the previous response to reviewers’ comments, the authors keep emphasizing that the novelty of this piece of work is the prepared LLZO/Polymer layer alleviating the reactivity of liquid electrolyte with Li anode by space charge layer/weakly coordinated Li ions in the “protective layer,” therefore enables stable cycling of LEAN electrolyte Li batteries. Although massive amounts of data are provided, the results do not sufficiently support the proposed theory. The cell design repeats previous publications since the decade on quasi-solid Li batteries in which a tiny amount of liquid electrolyte is used to wet the interfaces between electrodes and solid electrolyte.

RESPONSE: The authors very much appreciate Reviewer #2 for reviewing our manuscript and providing inspiring comments, and raising the detailed point out, which motivated us to clarify our concept. The previous publications on quasi-solid Li batteries in the solid electrolyte field in which a tiny amount of liquid electrolyte is used are focused on establishing intimate contact between the solid ceramic and the Li metal electrode, or using the liquid electrolyte as a plasticizer to facilitate the segmental motion of a polymer, i.e., to increase ion conductivity and reduce contact resistance. These approaches are intended to overcome the intrinsic limitations of solid electrolytes.

We appreciate Reviewer #2's feedback regarding the novelty of our work. While we acknowledge that the material and structure of the composite layer we used have been reported previously as a quasi-solid-state electrolyte, we demonstrated a novel working mechanism for reducing liquid electrolyte decomposition at the Li metal electrode by combining single-ion-conducting ceramic and single-ion-conducting polymer electrolytes. To the best of our knowledge, this mechanism has not been suggested and experimentally demonstrated before. Our findings represent a significant contribution to the field of liquid electrolyte-based Li metal batteries. As noted by both Reviewer #1 and the recent publications (*Nat. Energy* 2021, 6, 487, *Nat. Rev. Mater.* 2021, 6, 1036), reducing the consumption of liquid electrolytes in Li metal batteries is a critical issue, and our work provides a potential solution to this problem.

The reviewer’s concern about the main concept of this work is believed to have arisen because the main point described above was not sufficiently revealed in the main figure and introduction to the manuscript. Thus, we have revised **Fig. 1a** and its description to clarify the main point of this work as outlined below. We sincerely hope we can address any concerns

regarding the academic significance of our research compared to the previously reported cell designs.

Fig. R6: Schematic illustration of the effects of the S-CE/B-GE and S-CE/S-GE composite layers on the cycling stability of liquid electrolyte-based LMBs (which is identical to Fig. 1a).

(Lines 63–69)

We reveal that the composite layer of S-CE and S-GE (S-CE/S-GE) reduces the loss of liquid electrolyte on the Li metal electrode and improve the cycling stability of LMBs by changing the solvation environment of Li^+ in the layer (Fig. 1a); this effect is not seen with a composite layer comprising S-CE and a bi-ion-conducting polymer (B-PE)-based gel electrolyte (B-GE). Furthermore, we demonstrate that the S-CE/S-GE layer affects the morphology of Li anodes and the stability of the cathode–electrolyte interface in Li metal batteries.

To further support the proposed theory, we analyzed the coordination structure of the liquid electrolyte components in the two composite layers using Raman spectroscopy. The Raman spectra of the liquid-electrolyte-free composite layers (S-CE/B-PE and S-CE/S-PE), the electrolyte-swollen composite layers (S-CE/B-GE and S-CE/S-GE), and the liquid electrolyte (1.5 M LiFSI DME) were compared, as shown in Fig. R6. The signals from FSI^- (720 cm^{-1}) and DME solvent (from 830 to 890 cm^{-1}) enable the monitoring of the solvation structure of Li^+ , as FSI^- and DME are the components responsible for solvating Li^+ in the composite layers. The comparison between the S-CE/B-PE and S-CE/B-GE shows that the signal from FSI^- appeared after the electrolyte swelling, indicating that the FSI^- readily permeates through the S-CE/B-GE. In contrast, the S-CE/S-GE showed a much weaker FSI^- signal compared to the liquid electrolyte and S-CE/B-GE. This suggests that the mobile anions (FSI^-) hardly permeate

through the S-CE/S-GE layer owing to the presence of the fixed STFSI⁻ anions. The liquid electrolyte and S-CE/B-GE did not show any notable differences in the Raman signals from DME; the signal from free DME was more intense than that from coordinated DME. Remarkably, the S-CE/S-GE exhibited a stronger signal for free DME and a weaker signal for coordinated DME compared to the liquid electrolyte and S-CE/B-GE. The reduced intensities of FSI⁻ and coordinated DME suggest that the liquid electrolyte molecules are weakly coordinated with Li⁺ in the S-CE/S-GE composite layer.

We strongly believe that the Raman analysis provides solid evidence for the change in the solvation structure of Li⁺. Therefore, we added it to the revised manuscript as shown in **Fig. 2a**.

Fig. R6: Coordination structure of the liquid electrolyte components in the S-CE/B-GE and S-CE/S-GE. Raman spectra of the liquid electrolyte (1.5 M LiFSI DME), S-CE/B-PE, S-CE/B-GE, S-CE/S-PE, and S-CE/S-GE.

(Lines 150-166)

We analyzed the coordination structure of the liquid electrolyte components in the two composite layers using Raman spectroscopy. The Raman spectra of the liquid electrolyte-free composite layers (S-CE/B-PE and S-CE/S-PE), the electrolyte-swollen composite layers (S-CE/B-GE and S-CE/S-GE), and the liquid electrolyte (1.5M LiFSI DME) were compared, as shown in **Fig. 2a**. The signals from FSI⁻ (720 cm⁻¹) and DME solvent (from 830 to 890 cm⁻¹) enable the monitoring of the solvation structure of Li⁺ as FSI⁻ and DME are the components responsible for solvating Li⁺ in the composite layers. The comparison between the S-CE/B-PE and S-CE/B-GE shows that the signal from FSI⁻ appeared after the electrolyte swelling,

indicating that the FSI^- readily permeates through the S-CE/B-GE. In contrast, the S-CE/S-GE showed a much weaker FSI^- signal compared to the liquid electrolyte and S-CE/B-GE. It suggests that the mobile anions (FSI^-) hardly permeate through the S-CE/S-GE layer owing to the presence of the fixed STFSI⁻ anions in the layer. The liquid electrolyte and S-CE/B-GE did not show any notable differences in the Raman signals from DME; the signal from free DME was more intense than that from coordinated one. Remarkably, the S-CE/S-GE exhibited a stronger signal for free DME and a weaker signal for coordinated DME compared to the liquid electrolyte and S-CE/B-GE. The reduced intensities of FSI^- and coordinated DME suggest that the liquid electrolyte molecules are weakly coordinated with Li^+ in the S-CE/S-GE composite layer.

The difference in this draft is that the authors name the prepared (quasi)solid electrolyte layer (S-CE/S-GE) as a “protective layer” on the lithium surface, and the full cell then is LEAN electrolyte due to the small portion of liquid electrolyte. This is a misuse of the “lean electrolyte” concept. Otherwise, is this mean in an all-solid-state battery with only a dense LLZO layer to separate the cathode and metallic lithium anode, one can call it an electrolyte-free battery?

RESPONSE: The authors thank Reviewer #2 for raising these valid concerns. We fully understand and acknowledge that the term ‘lean electrolyte’ could be misleading to readers. Our intention was to convey that the mitigation of the electrolyte consumption by the S-CE/S-GE has a significant effect on the cell, particularly when only a tiny amount of liquid electrolyte is injected.

To address the reviewer’s concerns and avoid any confusion, the ambiguous expression ‘lean electrolyte’ was carefully rephrased throughout the entire manuscript. We use clear expressions such as ‘low liquid electrolyte content’ (referring only to liquid electrolytes) and ‘low mass of electrolyte content’ (including both the composite layer and liquid electrolyte). In addition, we present both the electrolyte-to-capacity ratio (E/C) of the injected liquid electrolyte (2.15 g Ah^{-1} , 1.28 g Ah^{-1}) and the E/C ratio including the weight of the composite layer (2.44 g Ah^{-1} , 1.57 g^{-1}) together. Even when including the solid electrolyte in the E/C ratio, it is a low value compared to previous studies.

Here, we evaluated the performance of a reference cell (Li||NMC) with a significantly larger amount of electrolyte (3.5 g Ah^{-1}) to verify the efficacy of the introduction of S-CE/S-GE. Our results showed that, even with an excessive amount of electrolyte, the performance of the

reference cell was significantly inferior compared to the cell containing S-CE/S-GE with an E/C of 2.44 g Ah⁻¹ (including the mass of the composite layer). We added the figure to our revised manuscript (**Supplementary Fig. 29**).

Fig. R7. Comparison of the cycling stability for the composite-layer-free Li||NMC cell with an E/C ratio of 3.5 g Ah⁻¹ and the S-CE/S-GE (E/C ratio of 2.44 g Ah⁻¹ including the mass of the composite layer).

In addition, as the previous reviewers mentioned, the presented battery performance is not outstanding. The Li battery with the complicated processing of a “protective layer” is not performing better than that of even using simpler coating compounds like LiF.

RESPONSE: We thank the reviewer for the kind reminder and appreciate the valuable feedback. First and foremost, we would like to clarify that our study proposes an interlayer strategy for Li metal cells with a low electrolyte content. It is important to note that most Li metal battery evaluations have been conducted in flooded conditions, where the electrolyte content exceeds 3 g Ah⁻¹, which is much higher than used in Li-ion batteries (~1.3 g Ah⁻¹). Therefore, it is essential to compare and consider not only cycle life but also electrolyte content in the cell to make a fair comparison.

In **Table R1**, the cell design parameters (cathode loading, N/P ratio, and E/C ratio) and the cycling stability are listed for the LiF coating strategies mentioned by the reviewer, the interlayer papers mentioned in the previous round of review in *Nature Energy*, and the studies on controlling liquid electrolyte reactivity through de-solvation. It should be noted that even in the comparison with the works not specifying the amount of electrolyte content or based on flooded conditions, the cell performance presented in this work is among the longest. Most

studies that achieve high performance with low electrolyte content adopted advanced electrolytes. As we showed in our previous manuscript (**Fig. 5g** and **Supplementary Table 2** in our previous manuscript), and as Reviewer #1 agreed, the performance we achieved with the conventional carbonate electrolyte is impressive.

Table R1. Comparison of the cycling stability, N/P ratio, and E/C ratio for our cell and previously reported pouch cells. Wave marks (~) indicate that we calculated the corresponding value for a paper that does not clearly indicate N/P or E/C.

		Reference	Cathode loading	N/P ratio	E/C ratio (g Ah ⁻¹)	Cycle number
1	Literature mentioned in the previous round of revision in Nature Energy	Adv. Funct. Mater. 2021, 31, 2006159	Not mentioned	Not mentioned	Not mentioned	120 or 300
2		Angew. Chem. 2020, 132, 2071–2076	8 mg cm ⁻²	8.57	Not mentioned	200
3		ACS Appl. Energy Mater. 2021, 4, 862–869	4.5 mg cm ⁻²	Not mentioned	Not mentioned	160
4		Adv. Mater. 2019, 31, 1808392	1 mAh cm ⁻²	8	~45 g Ah ⁻¹	150
5		Joule 3, 2761–2776, 2019	2 mAh cm ⁻²	4	~40 g Ah ⁻¹	160
6	Interlayer strategy with low electrolyte content	Fluorosulfonyl-DOL-GO layer (Nat. Mater. 2019, 18, 384)	3.4 mAh cm ⁻²	1.9	8.4 g Ah ⁻¹	200
7		Ag ₂ S layer (Adv. Energy Mater. 2022, 12, 2201390)	5 mAh cm ⁻²	2.17	0.92 g Ah ⁻¹	5
8		MoS ₂ layer (Chem. Electro. Chem. 2020, 7, 890)	4.2 mAh cm ⁻²	5.71	3 g Ah ⁻¹	170
9		UIO-66 nano-cage layer (Matter 2020, 3, 1685)	4 mAh cm ⁻²	1	2.76	70
10		Polar polymer host (J. Energy Chem. 2022, 64, 172)	4 mAh cm ⁻²	1.6	3	60
11	LiF coating strategies	LiF coating (1) (Energy Storage Mater. 2019, 16, 85)	1 mAh cm ⁻²	Not mentioned	~50 g Ah ⁻¹	200
12		LiF coating (2)	1.5–2.1 mAh cm ⁻²	Not mentioned	Not mentioned	200

		(Nat. Commun. 2019, 10, 900)				
13		LiF coating (3) (J. Energy Chem. 2019, 37, 197)	0.45 mAh cm ⁻²	444	~133 g Ah ⁻¹	100
14		LiF coating (4) (Adv. Energy Mater. 2022, 12, 2200337)	2.18 mAh cm ⁻²	~45.87	~22.9 g Ah ⁻¹	120
15		LiF coating (5) (J. Mater. Chem. A 2020, 8, 17229)	2.5 mAh cm ⁻²	40	Not mentioned	210
16	Controlling liquid electrolyte reactivity through desolvation	MOF coating (Energy Environ. Sci. 2020, 13, 4122)	9.4 mg cm ⁻² 1.5 mAh cm ⁻²	2.52	Not mentioned	400
				1	Not mentioned	100
17		MOF coating (Nat. Commun. 2022, 13, 172)	4.5 mg cm ⁻² 0.86 mAh cm ⁻²	93	Not mentioned	100
	This work		21.47 mg cm ⁻² 3.72 mAh cm ⁻²	2.15	2.15 g Ah ⁻¹	400
				2.15	1.28 g Ah ⁻²	100

The authors emphasize the less liquid electrolyte consumption in the cells with S-CE/S-GE layer is due to weakly coordinated Li ions in the “protective layer”. However, the experimental process was suggested differently. As Li metal chemically reacts with the solvents in liquid electrolytes (e.g., with EC <https://pubs.acs.org/doi/10.1021/acsaem.8b01707>), SEI layer will form and consume some electrolyte. When the lithium anode is coated with the S-CE/S-GE layer, a large amount of EC/DEC was introduced on the surface of lithium by tape casting. A layer of reaction products is already formed. This layer protects the lithium when it is in contact with liquid electrolytes. The manuscript has no result or discussion on such chemical aging influence. If the authors think the solvents used to tape cast the S-CE/S-GE layer are stable with metallic lithium, why not directly use a liquid electrolyte with such solvents in full cells? Besides, after coating the “protective layer”, the “free volume” of Li in contact with the liquid electrolyte in a battery is much smaller than bare lithium; of course, the consumption of liquid electrolyte is smaller in that case. Taking these points into account, the provided data barely support the mechanism proposed by the authors. The major chemical and physical influences are missing.

RESPONSE: We are very grateful for the reviewer's detailed comments on the potential factors affecting our experimental results, which include the effect of the casting solvent and the reduced free volume of Li. We fully recognized during the previous round of revision in *Nature Energy* that the comparison between the bare Li electrode and the electrode coated with S-CE/S-GE can be influenced by the preformed SEI during the composite layer coating or the reduced contact of the Li metal electrode with the liquid electrolyte components, as the reviewer mentioned. To rule out these effects, we manufactured S-CE/B-GE, which differs only in the single-ion conductivity of the polymers constituting the composite layer and compared it with S-CE/S-GE (Fig. 1a). The same procedure was used to prepare both the S-CE/B-GE-coated and S-CE/S-GE-coated Li electrodes.

Electrochemical analyses were performed to study the effect of S-CE/B-GE on electrolyte decomposition and we found that S-CE/B-GE did not significantly reduce electrolyte decomposition compared to the cell without the composite layer. This was evident from the linear sweep voltammetry (LSV) and chronoamperometry (CA) experiments (**Fig. 3b** and **c** in the revised manuscript), which showed similar currents for both cells. This indicates that neither the preformed SEI layer nor the reduced contact with the liquid electrolyte are the main factors for mitigating electrolyte decomposition. It should be noted that LSV and CA are experiments that observe electrolyte decomposition currents on the surface of Cu electrodes that do not react with slurry solvent. The authors thus concluded, based on the comparison between S-CE/B-GE and S-CE/S-GE, that the weak coordination of electrolyte molecules by the S-CE/S-GE layer is a key factor that influences the decomposition of the electrolyte.

The revised manuscript includes **Fig. R8 and R9** with a brief discussion on this, which the authors believe improves the comprehension of the experimental group for readers and rules out any artifacts occurring due to the reactivity between Li and the slurry solvent.

Fig. R8. Electrolyte reduction current via LSV in 1.5 M LiFSI DME + 50% TTFTE. (which is included in Fig. 3b).

Fig. R9. Electrolyte reduction current via LSV in 1 M LiTFSI EC/DMC + 3% LiBOB 10% FEC. (which is included in Supplementary Fig. 19).

(Lines 100–108)

Background to the introduction of S-CE/B-GE as a comparative group

To rule out the effects of reduced contact between the Li electrode and the liquid electrolyte by the ceramic and the polymer, or other incidental variables that occur in the slurry coating process, we compared the S-CE/S-GE with a composite layer with a bi-ion conducting polymer (B-PE)-based gel electrolyte (denoted as B-GE) consisting of lithium

bis(trifluoromethanesulfonyl)amide (LiTFSI) and PEGDA (see method section). The two layers, which were prepared in the same manner, both reduce the contact between the Li electrode and the liquid electrolyte. It should be noted that the Li^+ transference number is 0.61 for S-CE/B-PE and 0.96 for S-CE/S-PE, indicating a difference in ion transport property due to the polymer structure (**Supplementary Fig. 4**).

(Lines 231–240)

Discussion of LSV

We investigated how the weak coordination of Li^+ in the S-CE/S-GE affects the reductive decomposition of the electrolyte on the anode and consequent SEI formation. As illustrated in **Fig. 3a**, we set up three cells with different electrolyte states (liquid electrolyte, S-CE/B-GE, and S-CE/S-GE) between Li and Cu, and observed the reduction current of electrolyte molecules on the Cu surface. We first performed linear sweep voltammetry (LSV) for Li||Cu cell to observe the electrolyte reduction behavior. As shown in **Fig. 3b**, while the S-CE/B-GE shows similar behavior without the composite layer, the introduction of the S-CE/S-GE largely depressed the signals for anion and solvent reduction (highlighted by blue shadow). The lower reduction currents can be understood by diminished anion permeation into the S-CE/S-GE and decreased population of Li^+ -coordinated DME in the S-CE/S-GE.

Based on the adopted concepts and lacking novelty, considering the significance of this topic and the quality of evidence provided to explain the obtained phenomenon, I suggest rejecting the paper.

Revisions need to be made before submitting this paper to any other journals, including but not limited to the following aspects:

1. In figure 1b, the SEM image doesn't indicate the total thickness of SCL is 6-7 μm . The edge of the SCL is invisible.

RESPONSE: The authors appreciate the reviewer's comment. In our revised manuscript, the SEM image was replaced with a new image obtained by focused-ion beam that shows all edges of the cross-section (**Fig. R10**). The updated image presents a clearer and more distinguishable image of the cross-section, enabling readers to better observe the thickness of the composite layer.

Fig. R10. FIB-SEM images of the S-CE/S-PE composite layer coated on Li metal electrode (which is included in Fig. 1b).

2. The successful synthesis of S-PE is only proved by the IR spectrum in the supporting information. This is not enough since the CH₂ asym stretch is similar in all three samples. Besides, the IR result in red is clearly treated by baseline correction and so on, whereas the results of STFSILi and PSTFSILi were not. How can one compare the results with different treatments?

RESPONSE: The authors thank the reviewer for this comment, which raised a reasonable point regarding the C=C signal in our FTIR spectra. We did not perform a baseline correction for all the samples. The different baselines could be due to the different states of the samples (powder, flowing polymer, and crosslinked polymer). To clearly demonstrate the polymerization of STFSI and crosslinking by PEGDA, we have added an optical image of the synthesized polymers (**Fig. R11**). Additionally, we conducted a focused investigation of the C=C asymmetric stretching mode for the samples. As shown in **Fig. R12**, the signal of the C=C moiety disappeared for the P(STFSI)-co-PEGDA, completion of the polymerization reaction. We incorporated the figure into our revised manuscript (**Supplementary Fig. 1**).

Fig. R11. Optical images of (STFSI)Li, P(STFSI)Li, and P(STFSI)Li-co-PEGDA.

Fig. R12. FTIR spectra of (STFSI)Li, P(STFSI)Li, and P(STFSI)Li-co-PEGDA.

3. The liquid electrolyte used in symmetric cells differs from the one used in the full cell. How to correlate the characterization results done in symmetric cells to the full cell performance since they are different systems?

RESPONSE: We thank the reviewer for discussing the electrolyte system. The main figures of this study show the symmetric cell results with 1.5 M LiFSI DME + 50% TTFTE and the full cell results with 1 M LiTFSI EC/DMC + 10% FEC + 3% LiBOB, as mentioned by the reviewer. Since the ether electrolyte is a simpler electrolyte system used to show the working mechanism, its symmetric cell analysis was provided in the main figure. To demonstrate the practical applicability as a drop-in solution, the full cell result for the carbonate electrolyte was highlighted in the main figure.

In order to resolve the concerns raised by the reviewer regarding the differences according to the electrolyte system, we conducted all the analyses on the two electrolytes in this revision. For the carbonate electrolyte, Raman and MD simulation analyses revealed the formation of weak Li^+ coordination and enhanced interfacial transport property (**Fig. R13** and **R14**). LSV and CA analyses (**Fig. R15** and **R16**) experimentally confirmed the reduced electrolyte decomposition for the carbonate electrolyte, as they did for the ether electrolyte (**Fig. 3b** and **c**). Moreover, the investigation of the SEI chemical structure in the carbonate electrolyte indicated that the introduction of the S-CE/S-GE layer resulted in an increase of the F/C ratio and a decrease of the S/C ratio (**Fig. R17** and **R18**), similar to the ether electrolyte (**Supplementary Fig. 15** and **R16** in the original manuscript). It suggests the preferential involvement of the fluorinated solvent in SEI formation and the suppression of the anion

decomposition for both electrolytes. Furthermore, as observed for the carbonate electrolyte (Fig. 4a), the full cell cycling stability with the ether electrolyte was notably enhanced with the S-CE/S-GE (Fig. R19).

Fig. R13. Raman spectra before and after carbonate electrolyte permeation into the S-CE/B-PE and S-CE/S-PE. To distinguish anions in the liquid electrolyte and composite layer, LiFSI-based liquid electrolyte was used in this experiment.

Fig. R14. MD simulations with carbonate electrolyte. **a, b,** Number density profiles of Li^+ and electrolyte molecules at the interface of S-CE/B-GE (left) and S-CE/S-GE (right). The zero position corresponds to the outermost layer of the S-CE. **b,** Diffusivity of Li^+ in S-CE/B-GE and S-CE/S-GE structures. The diffusivity of Li^+ at the interface was calculated isotropically in the in-plane and through-plane directions. **c,** Contributions to the coordination number of Li^+ in B-GE, S-CE/B-GE interface, S-GE, and S-CE/S-GE interface.

Fig. R15. Electrolyte reduction current via LSV in 1 M LiTFSI EC/DMC + 3% LiBOB 10% FEC.

Fig. R16. Electrolyte reduction current via CA in 1 M LiTFSI EC/DMC + 3% LiBOB 10% FEC.

Fig. R17. Atomic F/C and S/C ratios of the SEI layers for bare Li and Li|S-CE/S-GE in 1 M LiTFSI EC/DMC + 10% FEC + 3% LiBOB.

In the SEI layer of Li|S-CE/S-GE, a high F/C ratio indicates a decrease in EC/DMC decomposition (weak organic by-products), and a decrease in the S/C ratio indicates a decrease in anion (TFSI⁻) decomposition.

Fig. R18. F1s and S2p spectra of the SEI layer in 1 M LiTFSI EC/DMC + 10% FEC + 3% LiBOB. a, Bare Li. b, Li|S-CE/S-GE.

In the SEI layer of Li|S-CE/S-GE, no anion decomposition signal was observed, whereas a decomposition signal for the fluorinated solvent was observed. This means that the fluorinated solvent, not the anion, contributes dominantly to the SEI.

Fig. R19. Li||NMC532 full cell performance with 1.5 M LiFSI DME + 50% TTFTE measured at 0.5 C constant current charging with 4.2 V cut-off voltage and 1.0 C discharging at 25 °C.

We rephrased the discussion on the design of the carbonate electrolyte and added all of the new analyses with the carbonate and ether electrolytes to the supplementary information. The experiments and analyses for the ether and carbonate electrolyte are summarized in **Table R2**.

(Lines 291–302)

We formulated a carbonate electrolyte that enables stable full cell operation, taking into account the role of the S-CE/S-GE. Our formulation utilized a blend of ethylene carbonate and dimethyl carbonate (EC/DMC), which is compatible with the NCM cathode. LiTFSI was adopted, which is relatively reduction-stable with Li metal thanks to the stable $-\text{CF}_3$ group^{23,36}. Additionally, we included fluoroethylene carbonate (FEC) as a co-solvent to form a SEI on the Li surface and lithium bis(oxalate)borate (LiBOB) as an additive to form a cathode–electrolyte interphase that prevents oxidative decomposition of the solvent or salt. The carbonate-based electrolyte also exhibited a weak coordination structure at the S-CE/S-GE interface (**Supplementary Figs. 17 and 18**), reduced electrolyte decomposition current (**Supplementary Figs. 19 and 20**), increased contribution of the fluorinated solvent to form SEI (**Supplementary Figs. 21 and 22**), and improved Li morphology and cycle performance (**Supplementary Figs. 23 and 24**).

Table R2. Experiments and analyses for the ether and carbonate electrolytes.

	Ether electrolyte	Carbonate electrolyte
Design concept	Main solvent: DME SEI former: TFTFE	Main solvent: EC/DMC SEI former: FEC
Weak solvation structure	Raman analysis (Fig. 2a, performed during revision) MD simulation (Fig. 3b, c)	Raman analysis (Supplementary Fig. 17, performed during revision) MD simulation (Supplementary Fig. 18)
Suppressed electrolyte decomposition	LSV experiment (Fig. 3b, performed during revision) CA experiment (Fig. 3c) XPS analysis (Supplementary Fig. 15, 16) Li morphology (Fig. 3d) Li symmetric cell analysis (Fig. 3e)	LSV experiment (Supplementary Fig. 19, performed during revision) CA experiment (Supplementary Fig. 20) XPS analysis (Supplementary Fig. 21, 22, performed during revision) Li morphology (Supplementary Fig. 23) Li symmetric cell analysis (Supplementary Fig. 24, performed during revision)
Applicability	Full cell test (Supplementary Fig. 25, performed during revision)	Full cell test (Fig. 4a)

4. Fig. 3e. The applied scale is so large that nothing is visible. E.g., what is the voltage difference between the cells? The cells should have different resistance. Why do they seem at the same potential? Besides, what is the reason for the potential increase for the cell with bare Li after 300 hours of cycling?

RESPONSE: The authors thank the reviewer for this helpful comment. We added an inset in the symmetric cell data (**Fig. R20**), which is identical to **Fig. 3e**, to aid readers in differentiating the overpotentials of the cells. Although the overpotential during the cell operation may vary due to various factors, such as surface roughening of Li and the building up of a porous layer, the initial overpotential tendency (1st–5th cycles) in our data was consistent with that by EIS (S-CE/S-GE < S-CE/B-GE < bare Li).

Regarding the overpotential increase for the bare Li, we measured the residual amount of Li source remaining in the Li electrode after the test. The results showed that the remaining Li source was approximately 2 mAh cm⁻² (equivalent to 10 μm Li, 25% of the initial thickness of 40 μm, **Fig. R21**), indicating that the increase in overvoltage was not caused by the depletion of the Li source. This result means that the observed increase in cell overpotential can be attributed to the accumulation of the porous layer and its insufficient wetting by the liquid electrolyte (*Nat. Energy* 2021, 6, 723).

Fig. R20. Bare Li, Li|S-CE/B-GE, and Li|S-CE/S-GE symmetric cell performance at 3 mAh cm⁻² and 5 mA cm⁻² (which is identical to **Fig. 3e**).

Fig. R21. Residual Li source measurements for fully degraded bare Li symmetric cell. The measurements were carried out by completely stripping the retrieved Li electrodes at 0.2 mA cm⁻².

5. The calculation and comparison with the literature shown in Fig. 5g and Table S2 seem unfair since the “protective layer” is excluded. Besides, the comparison is misleading for pouch cells in Table S2 due to the different cell assembly, cycling protocol and other unspecified vital factors.

RESPONSE: We appreciate the reviewer’s valuable comments. In response to the aforementioned comment, we included the weight of the protective layer in the E/C ratio presented in the revised manuscript. Regarding the comparison of pouch cell performance, the authors acknowledge the reviewer’s opinion that the direct comparison of the cell performance using only E/C ratio and cycle number in the main figure (**Fig. 5g**) could be misleading. In light of this comment, we removed **Fig. 5g** and **Table S2**, and carefully rephrased the related discussion in our revised manuscript.

(Lines 392–394)

The remarkable cycling stability of the full cell with the S-CE/S-GE composite layer, despite the E/C ratio of this work being the lowest value for LMBs with carbonate electrolyte^{5,19,41,43-50}, underscores the practical advantages of the S-CE/S-GE composite layer.

We would like to express our sincere appreciation to Reviewer #2 for providing us with insightful feedback that significantly improved the quality of our manuscript in terms of the main concepts, presentation, and experimental issues. The comments and suggestions from the reviewer were extremely valuable to us, and we believe that the revised manuscript adequately addresses the reviewer’s concerns. Once again, we thank the reviewer for taking the time to carefully evaluate our work and providing constructive feedback that helped to strengthen our study.

Reviewer comments, second round review:

Reviewer #1 (Remarks to the Author):

The authors have made tremendous efforts in addressing the comments raised. The new data on Raman spectra, the FIB examination of the Li thickness, and the completion of testing in both electrolytes have significantly improved the quality of the work. I recommend publication of the paper in its current form.

Reviewer #4 (Remarks to the Author):

This work presented a single-ion conduction composite protective layer on Li metal with a "lean" carbonate electrolyte to improve the cycling of the Li/NMC battery. It's a big challenge to have good cycling of Li metal battery with carbonate electrolyte at practical condition. The as-proposed protective layer on Li metal provides a good guidance to design high-energy-density Li metal battery. The ceramic layer on Li metal would enhance the battery safety as well. The revised manuscript enhances the understanding of the interaction of lithium ions with the composite layer according to the valuable suggestions of previous reviewers and well addressed the questions of the reviewers. After making some minor revisions, I would like to recommend this manuscript to be published in Nature Communications.

1. LiTFSI may have corrosive issue with Al foil at high voltage, is other lithium salt available to replace LiTFSI and the new salt will not affect the performance as well as change the Li+ coordination environment in composite protective layer?
2. It's well known that carbonate electrolyte is not the best electrolyte for Li metal battery, is as-proposed composite protective layer applied/compatible to other electrolytes? Like LHCE.
3. Are there any chances to further decrease the thickness/usage of composite layer? Like 2 microns. When it goes to large multiple layer pouch cells, the energy density will be affected by the thick composite layer.
4. In Table S1, the average voltage is 3.86V. It's too high for 4.3V NMC532. Please explain what's the average voltage. Please correct NMC811 in the Table S1 to NMC532.

Reviewer #1 (Remarks to the Author):

The authors have made tremendous efforts in addressing the comments raised. The new data on Raman spectra, the FIB examination of the Li thickness, and the completion of testing in both electrolytes have significantly improved the quality of the work. I recommend publication of the paper in its current form.

RESPONSE: We express our sincere gratitude to Reviewer#1 for conducting a thorough review of our revised manuscript. We are pleased to receive positive feedback from Reviewer #1 and appreciate their recommendation to publication.

Reviewer #4 (Remarks to the Author):

This work presented a single-ion conduction composite protective layer on Li metal with a “lean” carbonate electrolyte to improve the cycling of the Li/NMC battery. It’s a big challenge to have good cycling of Li metal battery with carbonate electrolyte at practical condition. The as-proposed protective layer on Li metal provides a good guidance to design high-energy-density Li metal battery. The ceramic layer on Li metal would enhance the battery safety as well. The revised manuscript enhances the understanding of the interaction of lithium ions with the composite layer according to the valuable suggestions of previous reviewers and well addressed the questions of the reviewers. After making some minor revisions, I would like to recommend this manuscript to be published in Nature Communications.

RESPONSE: We express our sincere appreciation to Reviewer#4 for dedicating their time and expertise to scrutinize our manuscript. We are pleased to receive recognition from Reviewer#4 regarding the effectiveness of our revision in adequately addressing the queries and apprehensions raised by the previous reviewer. We revised our manuscript according to the valuable questions and suggestions presented by the reviewer.

1. LiTFSI may have corrosive issue with Al foil at high voltage, is other lithium salt available to replace LiTFSI and the new salt will not affect the performance as well as change the Li⁺ coordination environment in composite protective layer?

RESPONSE: The reviewer questions the possibility of achieving weakly coordinated Li⁺ state with other Li salt that is free from Al corrosion issue. Although the imide-based salts are chemically stable in liquid electrolyte, they suffer from Al corrosion. Among the non-imide-based salts, LiPF₆ or LiBF₄ are known to be relatively free from Al corrosion (Nat. Mater. 21, 455-462, 2022).

We investigated the possibility of using LiPF₆ salt for the composite protective layer. As displayed in **Fig. R1**, the Li||NMC cell with a LiPF₆-based electrolyte (1 M LiPF₆ EC/DMC +10% FEC +3% LiBOB) exhibited an improvement in cycling stability when combined with the composite protective layer, demonstrating that the efficacy of the protective layer is not limited to LiTFSI.

However, the Al corrosion by LiTFSI can be mitigated by using an additive capable of forming cathode-electrolyte interphase like LiBOB. We found that the LiTFSI electrolyte with LiBOB additive has a higher cycling stability than LiPF₆ electrolyte in combination with the composite protective layer as shown in **Fig. R1**.

A brief mention on the Al corrosivity of LiTFSI was added in the revised manuscript.

Line (294~298)

LiTFSI was adopted, which is relatively reduction-stable with Li metal and chemically stable thanks to the stable -CF₃ group^{23,36}. Additionally, we included fluoroethylene carbonate (FEC) as a co-solvent to form a SEI on the Li surface and lithium bis(oxalate)borate (LiBOB) as an additive to form a cathode-electrolyte interphase that prevents oxidative decomposition of the solvent and aluminum corrosion by LiTFSI.

Fig. R1. Comparison of the cycling for the Li|S-CE/S-GE||NMC cell with the LiTFSI-based and LiPF₆-based electrolytes and the Li||NMC cell with the LiPF₆-based electrolyte.

2. It's well known that carbonate electrolyte is not the best electrolyte for Li metal battery, is as-proposed composite protective layer applied/compatible to other electrolytes? Like LHCE.

RESPONSE: Thank you for reviewer's inquiry regarding the compatibility of our proposed protective layer with state-of-the-art electrolytes for future applications. The recently

developed LHCEs or fluorine solvent-based liquid electrolytes has been designed to promote the FSI⁻ decomposition by increasing the Li⁺ coordination number of FSI⁻. It should be noted that our proposed single-ion conductor-based composite layer inhibits the transfer of salt anions and their decomposition at Li metal, thus interfering the SEI formation by FSI⁻. Considering the anion rejection by the composite protective layer, we employed a fluorine solvent capable of forming a stable SEI as an ingredient of the electrolyte.

As one of the electrolytes compatible with the composite protective layer, we proposed a LHCE electrolyte, 1.5 M LiFSI DME +50% 1,1,2,2-tetrafluoro-1-(2,2,2-trifluoroethoxy)ethane (TFTFE), in our manuscript. As shown in **Fig. R2 (Supplementary Fig. 25)** in the manuscript, the electrolyte demonstrate an improved LMB performance even when combined with the composite protective layer.

In sharp contrast, 1M LiFSI dimethylsulfamoyl fluoride (FSA, Energy Environ. Sci.13, 212-220, 2020) exhibited in a rapid performance degradation in performance in combination with the composite protective layer. It suggests that FSA hardly forms stable SEI layer when FSI⁻ is rejected from the interface.

These case studies indicate that our composite protective layer works well with an electrolyte that can form a stable SEI by not salt anion but free solvent (FEC or TFTFE). However, more intensive research is needed to fully understand the interplay between electrolyte and composite protective layer. A brief statement motivating the electrolyte design for the composite protective layer was added to the revised manuscript.

Line (409~411)

Based on the weakly coordinated Li⁺ structure due to the interplay between the liquid electrolyte and composite single-ion-conductor examined in this study, we expect that the efficacy of the composite single-ion conductor can be amplified by designing liquid electrolyte.

Fig. R2. Li||NMC and Li|S-CE/S-GE||NMC cell performance with 1 M LiFSI FSA or 1.5 M LiFSI DME +50% TTFE.

3. Are there any chances to further decrease the thickness/usage of composite layer? Like 2 microns. When it goes to large multiple layer pouch cells, the energy density will be affected by the thick composite layer.

RESPONSE: Owing to the relatively large size of ceramic particles, with a D50 value of 500 nm, achieving a uniform protective layer of 1-2 microns was challenging (**Fig. R3a**). Comparatively, a 2-micron thick protective layer exhibited a considerable decrease in performance due to its inhomogeneity (**Fig R3b**). Nonetheless, the decrease of the thickness from 6 to 2 micron leads to a marginal gain in weight-specific energy density by 2% and in volume-specific energy density by 3% (**Fig. R3c**), while substantially decreasing cell performance. Making the ceramic electrolyte particle smaller and more uniform would be needed to decrease the thickness of the protective layer, which is one of the future challenges.

Fig. R3. Morphology, symmetric cell performance, and projected energy density with 1 μm and 6 μm thickness protective layer. **a**, Top view and cross-section images of 1 μm and 6 μm thickness protective layers. **b**, Symmetric cell performance with 1 μm and 6 μm thickness protective layer. **c**, Projected energy density of Li|S-CE/S-GE||NMC532 with 1 μm and 6 μm thickness protective layer.

4. In Table S1, the average voltage is 3.86V. It's too high for 4.3V NMC532. Please explain what's the average voltage. Please correct NMC811 in the Table S1 to NMC532.

RESPONSE: Thank you for the reviewer's detailed review and comment. We committed a mistake in calculating the average voltage for discharge step. We included the charging voltages in the calculation of average discharge voltage, leading to a higher value than expected. We amended the mistake and revised Table S1 accordingly (Average voltage → Average discharge voltage, 3.86 V → 3.82 V).

Supplementary Table 1. Estimated gravimetric and volumetric energy densities of prototype pouch cells without and with S-CE/S-GE composite layer.

	Li NMC532	Li S-CE/S-GE NMC532	Li S-CE/S-GE NMC532
Al current collector (13 μm)	3.51 mg cm^{-2}	3.51 mg cm^{-2}	3.51 mg cm^{-2}
Cu current collector (8 μm)	7.17 mg cm^{-2}	7.17 mg cm^{-2}	7.17 mg cm^{-2}
Lithium metal (40 μm)	2.14 mg cm^{-2}	2.14 mg cm^{-2}	2.14 mg cm^{-2}
S-CE/S-GE layer (6 μm)	-	1.07 mg cm^{-2}	1.07 mg cm^{-2}
NMC532 cathode (61 μm)	21.47 mg cm^{-2}	21.47 mg cm^{-2}	21.47 mg cm^{-2}
PE separator (19 μm)	1.77 mg cm^{-2}	1.77 mg cm^{-2}	1.77 mg cm^{-2}
Electrolyte	E/C=2.15 g Ah^{-1}	E/C=2.15 g Ah^{-1}	E/C=1.28 g Ah^{-1}
Discharge capacity	3.72 mAh cm^{-2}	3.72 mAh cm^{-2}	3.72 mAh cm^{-2}
Average discharge voltage (0.1C)	3.82 V	3.82 V	3.82 V
Total stack energy (bi-cell)	28.41 mWh cm^{-2}	28.41 mWh cm^{-2}	28.41 mWh cm^{-2}
Total stack weight (bi-cell)	77.41 mg cm^{-2}	79.55 mg cm^{-2}	73.07 mg cm^{-2}
Stack gravimetric energy density	367.0 Wh kg^{-1}	357.1 Wh kg^{-1}	388.8 Wh kg^{-1}
Stack volumetric energy density	1088.5 Wh L^{-1}	1040.7 Wh L^{-1}	1040.7 Wh L^{-1}
Cycle number	<50 cycles	400 cycles	100 cycles